# Immunocompetent cell targeting by food-additive titanium dioxide

John W. Wills [1] ✉, Alicja Dabrowska [1], Jack Robertson[1], Michelle Miniter[1], Sebastian Riedle[2,3], Huw D. Summers[4], Rachel E. Hewitt [1], Adeeba Fathima[1], Alessandra Barreto da Silva [1], Carlos A. P. Bastos[1], Stuart Micklethwaite [5], Åsa V. Keita[6], Johan D. Söderholm [6], Nicole C. Roy [2,7,8], Don Otter[9], Ravin Jugdaohsingh[1], Pietro Mastroeni[1], Andy P. Brown [5], Paul Rees [4,10] & Jonathan J. Powell [1] ✉

Food-grade titanium dioxide (fgTiO$_2$) is a bio-persistent particle under intense regulatory scrutiny. Yet paradoxically, the only known cell reservoirs for fgTiO$_2$ are graveyard intestinal pigment cells which are metabolically and immunologically quiescent. Here we identify immunocompetent cell targets of fgTiO$_2$ in humans, most notably in the subepithelial dome region of intestinal Peyer's patches. Using multimodal microscopies with single-particle detection and per-cell / vesicle image analysis we achieve correlative dosimetry, quantitatively recapitulating human cellular exposures in the ileum of mice fed a fgTiO$_2$-containing diet. Epithelial microfold cells selectively funnel fgTiO$_2$ into LysoMac and LysoDC cells with ensuing accumulation. Notwithstanding, proximity extension analyses for 92 protein targets reveal no measureable perturbation of cell signalling pathways. When chased with oral $\Delta$aroA-Salmonella, pro-inflammatory signalling is confirmed, but no augmentation by fgTiO$_2$ is revealed despite marked same-cell loading. Interestingly, Salmonella causes the fgTiO$_2$-recipient cells to migrate within the patch and, sporadically, to be identified in the lamina propria, thereby fully recreating the intestinal tissue distribution of fgTiO$_2$ in humans. Immunocompetent cells that accumulate fgTiO$_2$ in vivo are now identified and we demonstrate a mouse model that finally enables human-relevant risk assessments of ingested, bio-persistent (nano)particles.

Titanium dioxide (TiO$_2$) is a highly versatile nano- and micro- particulate mineral with widespread use in consumer products and manufacturing processes, including as a food and pharmaceutical additive. Although, for many years[1,2], there have been questions over a potential role of ingested TiO$_2$ in initiating or augmenting an inflammatory response in the intestine, the global industry, valued at >\$17 billion per annum[3], is now facing a whole new bio-nano challenge. In May 2021, the European Food Safety Authority (EFSA) questioned the safety of

---

[1]Department of Veterinary Medicine, University of Cambridge, Cambridge, UK. [2]Riddet Institute, Massey University, Palmerston North, New Zealand. [3]Food Nutrition & Health Team, Food & Bio-based Products Group, AgResearch, Grasslands Research Centre, Tennent Drive, Palmerston North, New Zealand. [4]Department of Biomedical Engineering, Faculty of Science and Engineering, Swansea University, Swansea, UK. [5]School of Chemical and Process Engineering, University of Leeds, Leeds, UK. [6]Department of Surgery and Department of Biomedical and Clinical Sciences, Linköping University, Linköping, Sweden. [7]Department of Human Nutrition, University of Otago, Dunedin, New Zealand. [8]High-Value Nutrition Ko Ngā Kai Whai Painga National Science Challenge, Auckland, New Zealand. [9]Department of Food Science and Microbiology, Auckland University of Technology, Auckland, New Zealand. [10]Imaging Platform, Broad Institute of MIT and Harvard, Cambridge, MA, USA. ✉e-mail: jw2020@cam.ac.uk; jjp37@cam.ac.uk

food-grade TiO$_2$ (fgTiO$_2$; also termed E171) primarily citing unresolved genotoxicity concerns[4]. Based on this, European Commission member states agreed to remove fgTiO$_2$ as an additive in foods. This represented a seemingly pragmatic decision balancing need for this additive (e.g., aesthetics) against hypothetical risk. However, the European Commission also requested that the European Medicines Agency (EMA) evaluate the impact of removal of fgTiO$_2$ from the list of authorised excipients that are permitted in medicinal products. In this case, fgTiO$_2$ is not merely about aesthetics: it delivers opacity and protects actives from UV light degradation whilst aiding moisture control and patient compliance. The EMA concluded that any foreseeable ban, impacting the ~91,000 medicinal products that contain this pigment in Europe, risks causing pharmaceutical shortages[5]. Moreover, they estimated that a period of 7–12 years would likely be required for implementation of suitable alternatives[5]. Meanwhile, consumer concerns over the safety of fgTiO$_2$ have been expressed beyond Europe. Notably, in 2022, a class action suit was filed against the multinational food company Mars Incorporated in the USA alleging that consumers of the hard-shell candy Skittles are at heightened risk of health effects stemming from genotoxicity[6].

Whilst regulators in other continents / territories have not followed EFSA's 2021 decision, there is widespread agreement that the available data addressing the uptake, biological fate and effects of fgTiO$_2$ are insufficient[7–9]. In fact, direct toxicity of fgTiO$_2$ in cell culture experiments has not been consistently demonstrated, especially when conditions are carefully controlled to avoid artefactual in vitro cell gorging[10]. In contrast, it is recognised in vitro that fgTiO$_2$ can modestly augment cellular responses to primary danger signals (e.g., microbe-associated molecular patterns) although cell loading of the particles needs to be relatively high[1]. Similarly, Kirkland et al.[11], identified any potential genotoxic activity of various titanium dioxide polymorphs as secondary to cell stress which, certainly for fgTiO$_2$, requires marked cell loading[1].

Aside from specific cells of the small intestine, there are no known circumstances under which fgTiO$_2$ markedly accumulates in vivo. Systemic exposure to fgTiO$_2$ appears to be very low[12]. Even allowing for lifetime accumulation in humans it appears that anticipated target organs (liver and spleen) contain levels of Ti and Ti-rich particulates that are just above the tissue-free background, and the origin of this slightly enhanced signal is unknown[13]. Consistent with this, many animal feeding studies show almost no detectable systemic accumulation of fgTiO$_2$[12,14] and, overall, there are no convincing ex vivo imaging studies showing multiple, heavily-loaded cells. The exception, as noted above, is the small intestine, in both humans and rodent models[15,16], and specifically their large lymphoid follicles termed Peyer's patches. In absence of afferent lymphatics, the apical surfaces of these follicles possess a unique population of microfold (M) cells that avidly sample the lumen for bulky macromolecules including small micro- and large nano- particles (herein referred to as particles). This material is passed to underlying immune cells in the apical, subepithelial dome (SED) region of the follicle which contains recognised populations of monocyte-derived, lysozyme-expressing cells termed LysoMacs and LysoDCs that are specialised for the uptake of dead cells, particulate antigens and pathogenic bacteria, in addition to their role in antimicrobial defence[17]. Overall, the major role of Peyer's patches is initiating processes for the generation of mucosal IgA and systemic IgG in response to intestinal antigen thereby orchestrating the sequence of events that leads to gut-derived humoral immunity. Over time, fgTiO$_2$ is known to accumulate in Peyer's patches, alongside other persistent particles, in basally located tissue-fixed macrophages termed pigment cells[15,16]. These cells are of low immunological and metabolic activity and appear to act as lifetime graveyards for persistent particles, presumably limiting systemic absorption through local sequestration in the same way that tissue-fixed macrophages sequester particulate tattoo inks[15,16,18,19]. So the paradox remains: whilst fgTiO$_2$ is under intense regulatory scrutiny, there is no known site of accumulation in active (e.g., immunocompetent) cells that could plausibly give cause for concern or that allows informed risk assessment.

In this work, we address whether similar, selective cell loading occurs elsewhere (i.e., away from the safe house base-of-patch pigment cells) and what cells are targeted, at what frequency and with what impact. The approach uses quantitative image analysis of light and electron microanalytical data, correlative mouse-human cell dosimetry and broad-target interrogation of fgTiO$_2$'s capability to augment a primary cellular danger signal in vivo.

## Results

### Titanium dioxide in human intestinal tissues

High-resolution reflectance confocal microscopy of human small intestinal tissue samples identified bright foci at the base of Peyer's patches (Fig. 1a) consistent with previous reports of pigment cells in this region[16,18,20]. These cells are known to mostly accumulate ingested engineered aluminosilicates and fgTiO$_2$[16,20]. Using Al and Ti as biomarkers for these particles, correlative analyses using scanning electron microscopy (SEM) and energy dispersive X-ray spectroscopy (EDX) confirmed the presence and co-localisation of these elements (Fig. 1b; top row). Reflectant foci were found entirely attributable to Ti-rich particles, as areas with Al but no Ti signals were not reflectant, whereas all detectable Ti foci yielded reflectance (Fig. 1b; rows 2–5). Alongside further checks using mouse tissues known to contain fgTiO$_2$ or not (shown, Supplementary Fig. 1), the reflectance imaging approach was confirmed as a robust, high-throughput means to detect fgTiO$_2$ in tissue[15].

Of significant note, previously unreported populations of human cells containing similar, reflective material were also identified (i) sporadically in the regular lamina propria (Fig. 1c) and (ii) densely in the immuno-active subepithelial dome (SED) region of Peyer's patches (Fig. 1a). These cell regions were also subjected to correlative SEM-EDX analyses and, again, reflectant foci were a match for the Ti signal (Fig. 1d/e). Given the unexpected density of the identified Peyer's patch population, whole SED regions from a randomly selected series of nine human subjects were imaged using the analytically-validated confocal microscopy approach. Albeit with marked inter-individual variability, reflectant signal was found in the SED of every human case (Fig. 1f). Importantly, the recipient cells often exhibited multiple foci per-cell (i.e., high cellular loading), suggesting specific cellular targeting and accumulation of fgTiO$_2$ (Fig. 1f – insets) in a non-pigment cell population.

### Recapitulation in a mouse model

To investigate the pathways giving rise to the above observations, and to phenotype the population of fgTiO$_2$-accumulating cells, in Mouse Study 1, we turned to a recently reported murine model for human-relevant, oral exposure to fgTiO$_2$[15]. Initially we tilescanned a complete ileal transverse section using high-resolution confocal reflectance microscopy and used image analysis to segment all cells (cell segmentation accuracies assessed, Supplementary Fig. 2). This allowed the cellular dose of fgTiO$_2$ to be measured in each cell-object (i.e., the integrated intensity of the thresholded reflectance signal per cell) enabling the subdivision and display of the cellular loading of fgTiO$_2$ in three groups: low, intermediate and high (Fig. 2a). As in the human samples (Fig. 1), the patch region was decorated with reflectance signal, notably in the SED and at the base, and cellular loading was often high – especially in the SED (Fig. 2a). However, unlike in human tissue, signal was entirely absent away from the patch except for two small, isolated foci in the villous epithelium. The resolution of the imaging process allowed detailed investigation of these latter events confirming that this signal was not inside the tissue, but derived from material within goblet cell invaginations that were open to the intestinal lumen (Fig. 2a–inset).

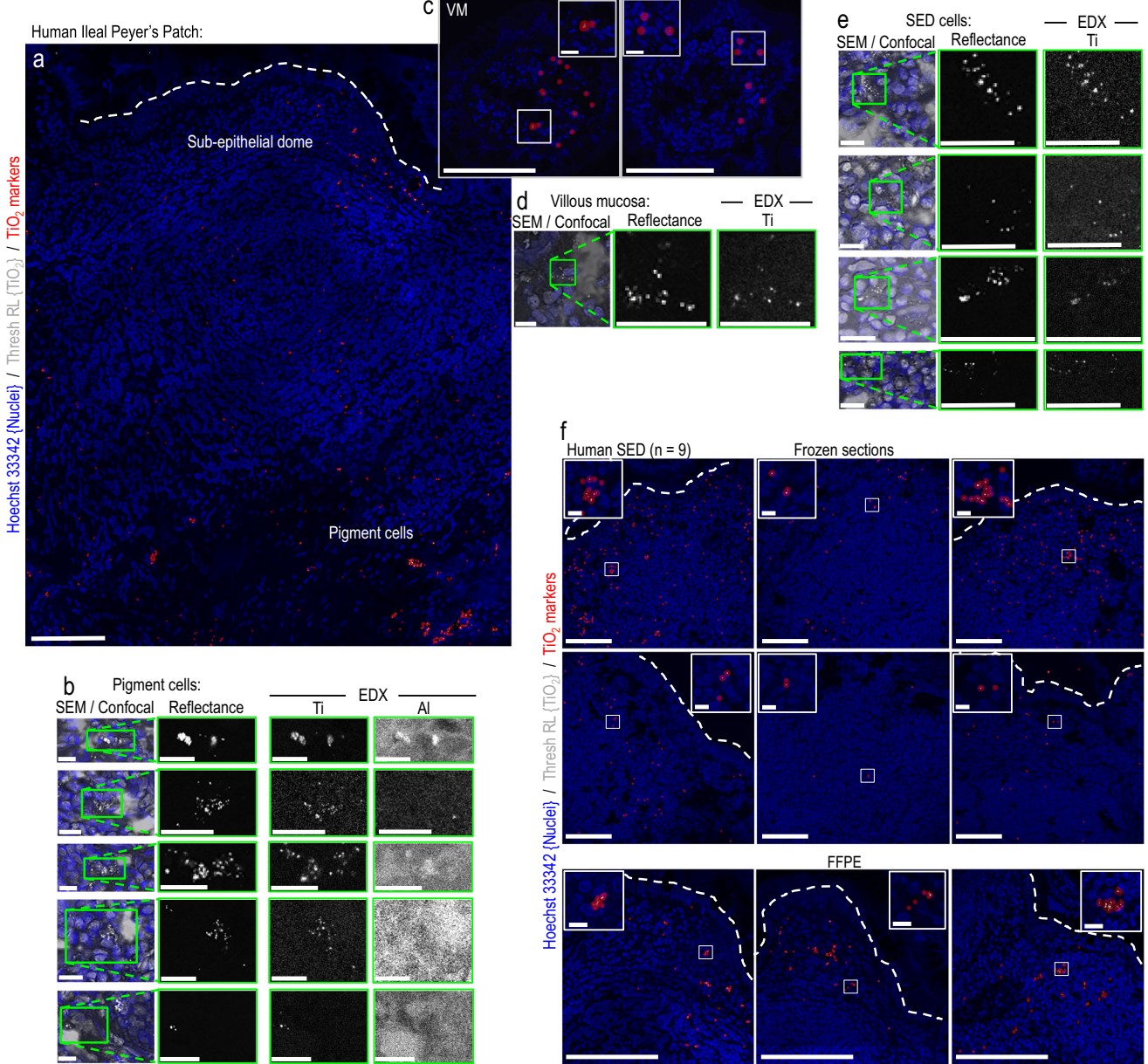

**Fig. 1 | Titanium dioxide in human intestinal tissues. a** Tilescanned image acquired by confocal reflectance microscopy. Reflectant foci were detected along the follicle base consistent with previously-reported mineral particle-containing pigment cells. Of note, similar reflectant foci were also present apically in the immuno-active, subepithelial dome tissue-region, and away from the patch in the (**c**) lamina propria of the regular villous mucosa (VM). Translucent red circle-markers are placed on the reflectant foci to aid visualisation. **b**, **d**, **e**, Correlative SEM/EDX analyses performed on the same tissue section shown in (**a/c**). The pigment cell region is shown in (**b**) where X-ray signal for both Al (attributable to aluminosilicates) and Ti (attributable to titanium dioxide) was found. EDX analyses in the (**d**) villous mucosa and (**e**) subepithelial dome regions showed reflectant foci were attributable to Ti. **b**, **d**, **e** In all instances the elemental data for Ti were a near-exact match for the reflectant foci, whereas (**b**) areas with Al signal alone did not reflect. The image-data presented in (**a**–**e**) are representative of independent repeats in $n = 2$ samples. **f** Reflectance microscopical analyses of nine, randomly-drawn human samples (six frozen, three formalin-fixed paraffin embedded (FFPE)) showed reflectant foci consistent with $TiO_2$ in the subepithelial dome of every sample. Scale bars: (**a**) = 50 μm; (**b**, **d**, **e**) = 10 μm; (**c**, **f**) = 50 μm with 10 μm insets.

To confirm the apparent selectivity of $fgTiO_2$ for the lymphoid follicles—rather than the regular villous mucosa—we questioned whether a single particle in intestinal tissue would be missed by our reflectance microscopy approach. To do this, a tissue region containing a single, small yet clearly visible reflectant foci was milled out under correlative SEM, using a focussed ion beam (Fig. 2b–d). The milled-out lamella was then transferred to a transmission electron microscope for imaging with analysis by EDX and electron diffraction. A single particle of $fgTiO_2$, with a diameter around 100 nm and an anatase diffraction pattern consistent with dietary exposure, was identified (Fig. 2e–i). Since single, disperse particles of $fgTiO_2$ were therefore detectable by

the reflectance imaging strategy, we could confirm that ingested $fgTiO_2$ had not been missed with our imaging approach in the regular villous mucosa and thus was not taken up in this region (Fig. 2a).

Following from this, we next considered the caecal patches (i.e., large lymphoid follicles of the first part of the large intestine that adjoins the terminal ileum where Peyer's patches are located). Here, transverse tissue sections revealed almost no reflectant foci showing that, unlike Peyer's patches, the neighbouring caecal patch is spared from particle targeting (Fig. 2j). In spite of the marked similarity of these secondary lymphoid sites, the microfold (M) cell density of their overlying epithelium differs greatly: caecal patches have very few (like

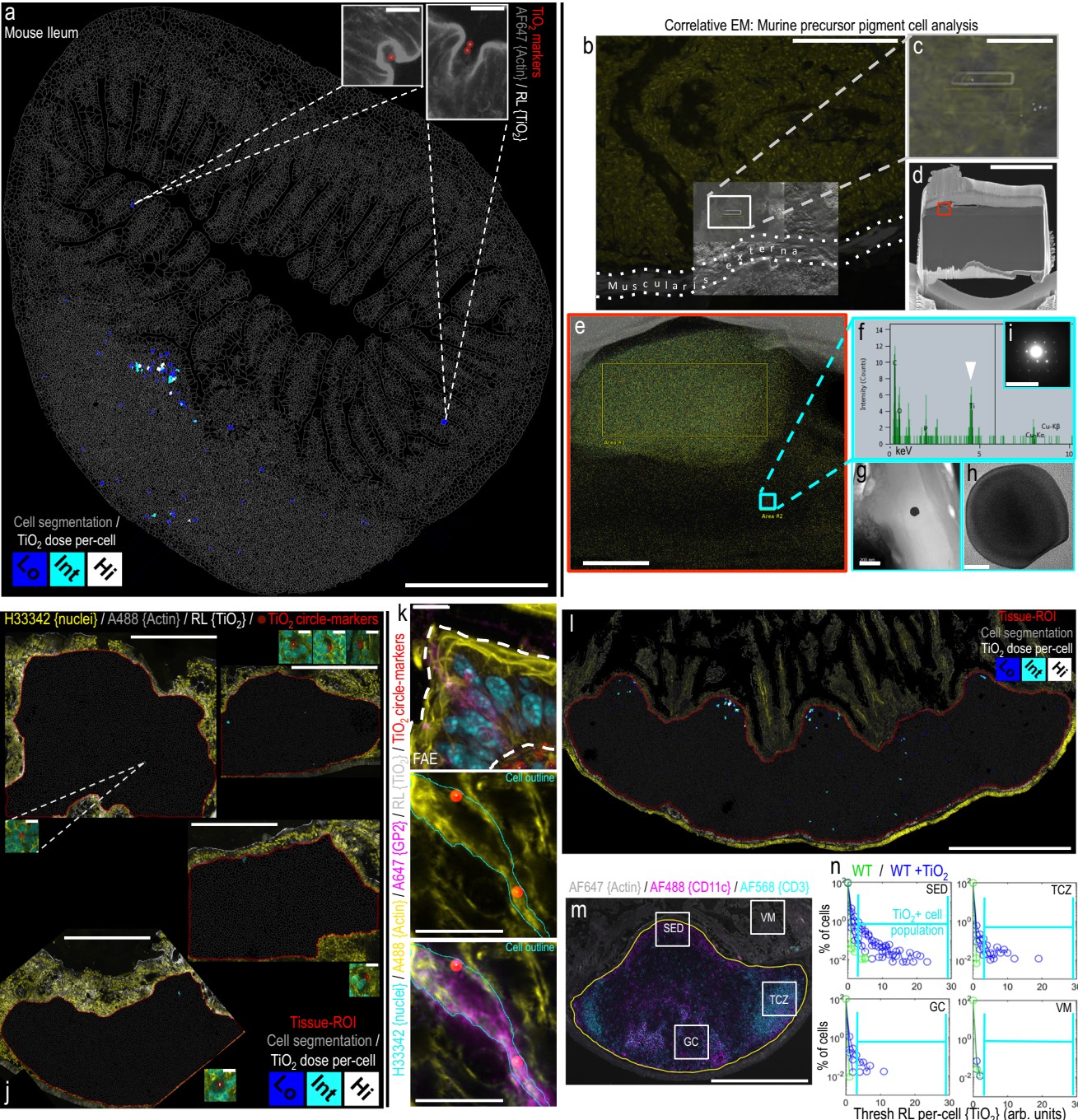

**Fig. 2 | A mouse feeding study demonstrates fgTiO₂ specificity for Peyer's patches. a** Single-cell analysis of a tilescanned ileal tissue section. As in humans (Fig. 1a/f), fgTiO₂ loaded into the subepithelial dome (SED) and there was evidence of precursor pigment cell formation along the follicle base. Beyond the follicle, two positive cells (**a**, insets) were found in the villous mucosa. In contrast to the human findings (Fig. 1c) both were caused by luminal fgTiO₂ trapped in the invaginated space of goblet cells. **b**–**d** To investigate the sensitivity of fgTiO₂ detection by confocal reflectance microscopy, a tissue-region containing a single, reflectant foci was milled out under correlative SEM. **e**-**i** Transfer of the lamella to a transmission electron microscope enabled imaging and X-ray (EDX) and electron diffraction analysis. A (**g**, **h**) single particle of fgTiO₂ with an (**i**) anatase diffraction pattern (i.e., as added to the diet) was confirmed responsible for the single reflectant foci observed. **j** Caecal patches (adjacent at the top of the colon) were almost completely devoid of fgTiO₂ (n = 4 animals). **k** Immunofluorescence microscopy

showed fgTiO₂ access to Peyer's patches was via GP2-positive microfold (M) cells. **l** Single-cell analysis of a longitudinal ileal section consistently showed the close association between an M-cell rich, follicle-associated epithelium and fgTiO₂ uptake. **m**, **n** Guided by CD11c and CD3 labelling for phagocytic mononuclear cells and T-lymphocytes (respectively) image-sets were collected from the SED, germinal centre (GC) T-cell zone (TCZ) or overlying villous mucosa (VM) tissue regions (n = 4 mice per diet-group). Some fgTiO₂ signal was observed in the GC and TCZ regions but the majority was in the SED. In keeping with (**a**), and in contrast to the human findings (Fig. 1c), in all four mice, no uptake was seen in the villous mucosa demonstrating specificity for M-cell mediated, Peyer's patch targeting. Scale bars: **a** = 500 μm with insets 5 μm; **b** = 50 μm; **c** = 10 μm; **d** = 5 μm; **e** = 500 nm; **g** = 200 nm; **h** = 20 nm; **i** = 10 1/nm; **j** = 500 μm with insets 10 μm; **k** = 10 μm; **l**, **m** = 500 μm.

the villous epithelium) with little capacity for particle uptake, whereas Peyer's patch epithelium is enriched with mature M-cells that exhibit distinct particle uptake capability for small particulates in general[21,22]. Indeed, cells of the Peyer's patch epithelium stained positively for the mature M-cell marker (GP2) and, here-and-there, these M-cells were caught in the act with co-signal for fgTiO$_2$ (Fig. 2k) as had long been assumed for fgTiO$_2$ but not actually demonstrated[18]. Indeed, longitudinal sections of ileal tissue containing Peyer's patches consistently demonstrated the close association between fgTiO$_2$ signal and an overlying, M-cell rich, follicle-associated epithelium (Fig. 2l). In this way, the above findings provide a rationale for the specificity of fgTiO$_2$ uptake by the Peyer's patches of the small intestine. It would appear, therefore, that to explain the presence of fgTiO$_2$ in the human intestinal lamina propria a mechanism other than direct uptake by the regular villous epithelium must exist (this is addressed below in Mouse Study 2). First, however, we quantified how cell loading with fgTiO$_2$ varied amongst the major different functional zones of the Peyer's patch.

Sets of confocal reflectance images were collected from the SED, germinal centre (GC), T-cell zone (TCZ) and overlying villous mucosa (VM) tissue regions (indicated, Fig. 2m) using tissue sections taken from four mice fed with the fgTiO$_2$-supplemented diet (+fgTiO$_2$), alongside four control mice from the same experiment and ingesting the same diet but without added fgTiO$_2$. Single-cell analysis, using flow cytometry-type gating informed by the negative control tissues, demonstrated some fgTiO$_2$ signal in the TCZ and GC but established the dominance of the signal and the highest cell loading in the biologically-active SED (Fig. 2n). As per the whole ileal cross-section data (Fig. 2a), in all sections analysed across the four mice, the villous mucosa was entirely devoid of fgTiO$_2$-positive cells and consistently exhibited the same reflectance distribution as the negative control mice (Fig. 2n).

### Establishing correlative human-mouse dosimetry

Rodent models provide pragmatic access to healthy pristine tissue (i.e., rapidly flash frozen at necropsy) from fully controlled experiments (i.e., population-matched with availability of fgTiO$_2$-negative tissues) in a way that human surgical specimens do not. Nonetheless, it is critical that a murine feeding model appropriately reports on the human situation. Next, therefore, we exploited the window of insight provided by precision image analysis to consider how the cellular dosimetry of fgTiO$_2$ in the SED cells of the murine model reflected SED-cell exposure in humans. To do this, the snap-frozen Peyer's patch tissue sections, taken from six randomly-drawn human samples, were compared with six murine samples taken from the fgTiO$_2$-fed group. Visually, in both species, there was marked inter-individual variability in fgTiO$_2$ concentration in the SED (Fig. 3a–l). To quantify this, we analysed the thresholded reflectance signal in the tissue sections in four ways, namely: thresholded reflectance per unit tissue area (i.e., fgTiO$_2$ abundance; Fig. 3m), thresholded reflectance per cell (i.e., fgTiO$_2$ load per cell; Fig. 3n), the number of reflectant foci per cell (i.e., cellular count of fgTiO$_2$-loaded vesicles; Fig. 3o), and finally the amount of reflectance signal per foci (i.e., fgTiO$_2$ load per vesicle; Fig. 3p). The strategy is schematically explained in Supplementary Fig. 3; cell segmentation accuracy assessments are presented in Supplementary Fig. 2; background reflectance comparisons in mice and humans are shown in Supplementary Fig. 4 and examples of the raw reflectance signal, resulting circle-marker placements and fgTiO$_2$-loaded vesicle segmentations are shown in Supplementary Fig. 5. By all metrics, the data from the two groups showed remarkable overlap and under statistical comparison none of the sets of measured fgTiO$_2$ distributions were found to be entirely unique to either species (presented, Supplementary Fig. 6). Notably when Peyer's patch loading of fgTiO$_2$ from mice on lower-dose-fgTiO$_2$ diets was analysed, the cellular accumulation of particles in the SED was several orders of magnitude

lower compared to the human situation (Supplementary Fig. 7). The above findings therefore establish the apparent ability of the murine model to provide human-relevant data—quantitatively as well as qualitatively—in terms of fgTiO$_2$ cell targeting and dosimetry in the biologically-active SED region of the Peyer's patch.

### fgTiO$_2$ selectively targets immunocompetent cells

The fgTiO$_2$-targeted cells in the SED were, unsurprisingly, CD11c$^+$ (i.e., phagocytic mononuclear cells) and not only was the frequency of such cells unaffected by the presence of fgTiO$_2$ particles so was their expression of the CD11c integrin biomarker (Fig. 4a–h). Neighbouring cells were chiefly CD3$^+$ (T cells) and B220$^+$ (B cells) and, similarly, their expression patterns and frequencies were not significantly affected by the presence of fgTiO$_2$ (P ≥ 0.26, two-sided unpaired samples T-tests) (Fig. 4a–h). Within the Peyer's patch, CD11c$^+$ cells are known to exhibit a number of sub-phenotypes in distinct follicular locations[17]. In this regard, the SED localisation and autofluorescence of the fgTiO$_2$$^+$ cells were entirely consistent with LysoMacs and/or LysoDCs which represent previously-described macrophage and dendritic cell subsets, respectively, that express high levels of lysozyme alongside a unique autofluorescence signature[17,23] (see Fig. 4i–s and Supplementary Fig. 8 for the autofluorescence spectra collected). Indeed, phenotypic markers that help differentiate LysoDCs from LysoMacs (cell surface CD4 and MHCII) demonstrated a mixed population of these two cell types for the fgTiO$_2$$^+$ cells (Fig. 4t/u).

Whilst similar populations of autofluorescent$^{hi}$ SED cells were observed in all mice regardless of diet, these cells were strongly particle-positive in mice receiving the dietary fgTiO$_2$ (Fig. 4m/s, Supplementary Fig. 9 and Supplementary Movie 1). Upon quantification (Fig. 4o), only ~7% of autofluorescent cells showed no clear particle signal for the +fgTiO$_2$ group (Fig. 4i–s). These cells are known to be important in the initial uptake and processing of luminal antigen which other work suggests may be coated in calcium phosphate to facilitate enzyme resistance during the early phases of the lumen-to-phagocyte sampling process[24]. Except in Crohn's disease, these cells have been shown previously to be positive for the immunoregulatory cell surface ligand PD-L1[25]. Again, consistent with this prior work, our quantitative single-cell tissue analyses showed that the fgTiO$_2$$^+$ (and therefore CD11c$^+$, autofluorescence$^{hi}$) cells of the SED were PD-L1$^+$ (Fig. 4v–y, antibody controls presented Supplementary Fig. 10). This contrasted with precursor pigment cells at the base of the Peyer's patch which were PD-L1$^{lo}$, despite their fgTiO$_2$ content (Fig. 4v–z). In other words, based on this finding, and that mice unexposed to fgTiO$_2$ show similar PD-L1$^+$ distributions (Fig. 4w), we can conclude that expression of this immunoregulatory molecule is unaffected by fgTiO$_2$ uptake (P = 0.34, z = −0.96, two-sided Wilcoxon rank-sum analysis) and, rather, fgTiO$_2$ is selectively and specifically shuttled into already-PD-L1$^+$ LysoMacs/LysoDCs in the SED of Peyer's patches in the small intestine. As such, we show that there is an immunocompetent cell population that accumulates fgTiO$_2$ in humans and in a rodent model, which will now allow the proposed adverse effects of the particulates to be tested.

### Potential for fgTiO$_2$−Salmonella interactions

As noted above, in earlier in vitro work, we have shown that the phagocyte proinflammatory response can be augmented by cellular loading with fgTiO$_2$[1]. As such Mouse Study 2 considered the potential for fgTiO$_2$ to initiate or augment an inflammatory response, precisely in the Peyer's patch-rich, terminal ileal region where the particles accumulate. Initiation was measured directly whilst for augmentation, we investigated the synergy between fgTiO$_2$ and an orally-delivered auxotrophic Salmonella strain, namely ΔaroA-deficient Salmonella enterica serovar Typhimurium (ΔaroA-Salmonella). The choice of bacterial infection was because (i) again, it is human-relevant (i.e., commonly experienced[26]) (ii) it targets LysoDCs and LysoMacs[27] resulting in an inflammatory response (unlike bacterial fragments, to

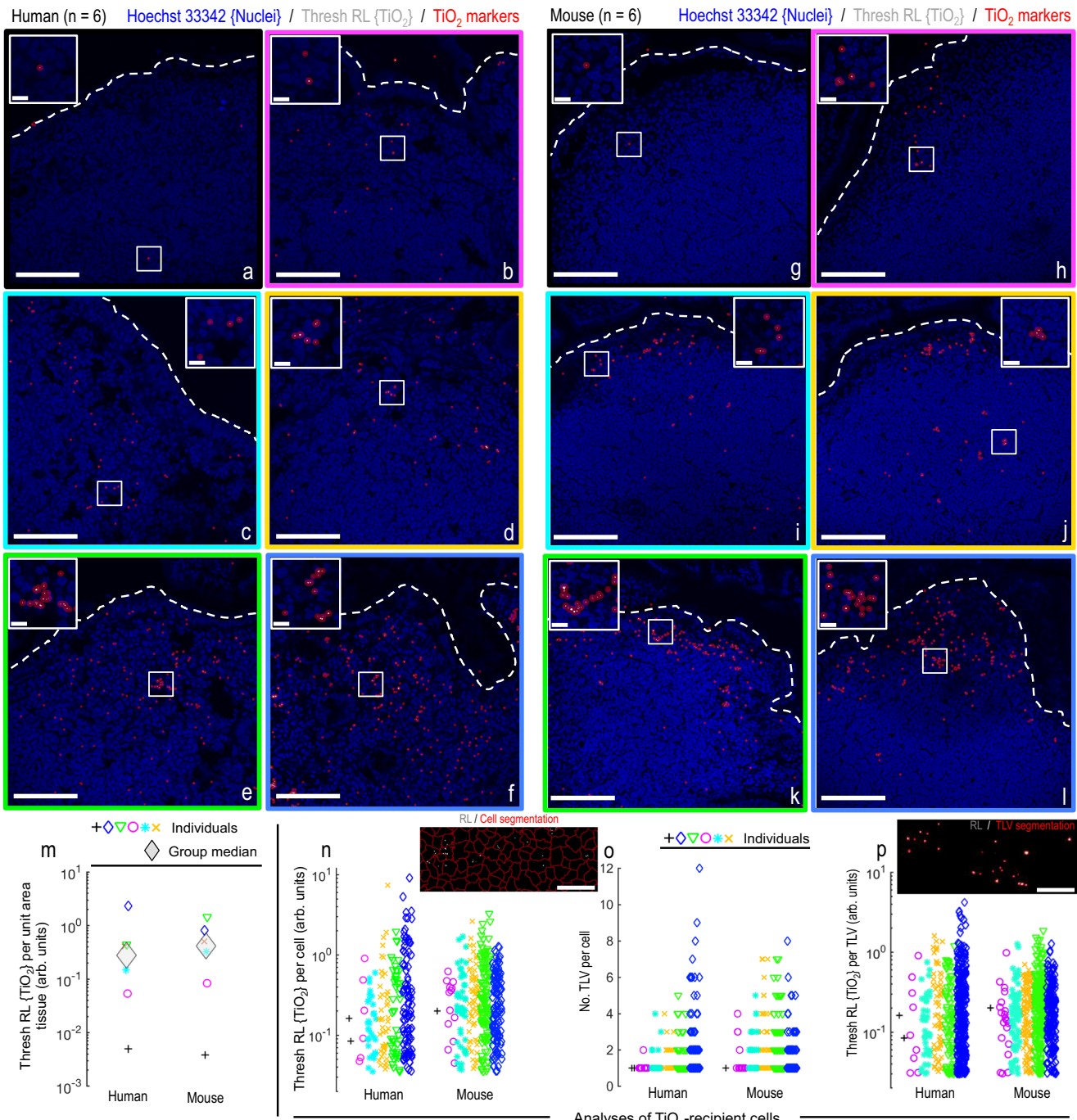

**Fig. 3 | Establishing correlative human-mouse dosimetry. a–l** Confocal reflectance microscopy showing the range of cellular loading in the subepithelial dome tissue region of (**a–f**), six randomly-drawn human and (**g–l**) six randomly-drawn mouse samples. After acquisition, the image-fields per specimen were manually laid out in order of lowest-to-highest fgTiO$_2$ SED accumulation to visually present the wide variation in fgTiO$_2$ cellular loading. Translucent red circle-markers were placed on thresholded reflectant foci to aid visualisation. **m–p** Quantitative analysis of the image-data shown in (**a–l**) ($n = 6$ mice / $n = 6$ humans). **m** Thresholded reflectance per unit tissue area (i.e., amount of fgTiO$_2$ per unit tissue area).

**n** Thresholded reflectance per cell (i.e., fgTiO$_2$ dose per cell). **o,** Number of thresholded reflectant foci per cell (i.e., number of fgTiO$_2$-loaded vesicles (TLV) per cell). **p** Thresholded reflectance signal per foci (i.e., fgTiO$_2$ dose per TLV). Statistical comparison of the distributions is presented in Supplementary Fig. 6 (two-sided Wilcoxon rank-sum analysis). None of the sets of measured fgTiO$_2$ distributions were found to be entirely unique to either species. **m–p** By all of the quantitative measures established, feeding a murine diet supplemented with 0.0625% (w/w) fgTiO$_2$ for eighteen weeks provides significant overlap with measured, real-world human exposures. Scale bars: **a–l** = 50 μm with insets 10 μm; **n** = 20 μm; **p** = 5 μm.

which the intestine is uniquely tolerant) and (iii) this response can be augmented leaving headroom for any interactions to be observed[28,29]. Regarding this latter point, it is known that same-cell concurrence of an amplification signal, such as over-expressed bacterial flagellin or co-expressed vectorised cytokines, is required to augment the *Salmonella*-induced immuno-inflammatory response[29,30]. For this reason, we

first demonstrated substantial same-cell occurrence of the particles and bacteria, in the SED of Peyer's patches, when fgTiO$_2$-fed mice (for 16 weeks) were immediately chased with a single oral dose of *ΔaroA*-*Salmonella* and tissues collected 3 days later (+3 days; Fig. 5a-c). *Salmonella*-specific faecal IgA was undetectable at this time-point but was clearly demonstrated in a second group at +28 days following the

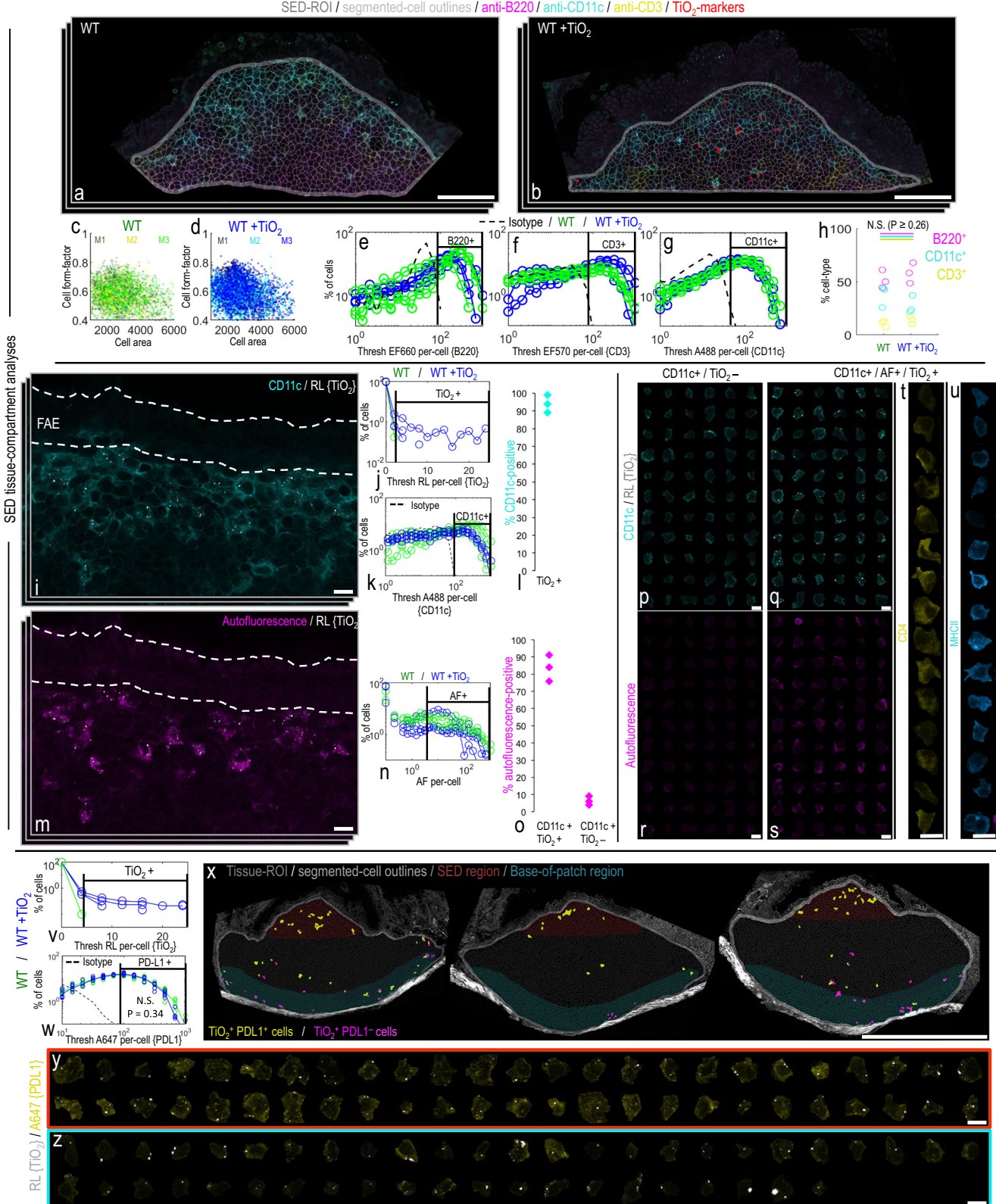

*Salmonella* challenge. This pattern was irrespective of fgTiO$_2$ ingestion (Fig. 5d) and is consistent with a latent immuno-inflammatory response following oral exposure to the pathogen[31,32].

To investigate both initiation and augmentation of an inflammatory response, and to cover a broad array of potential outcomes, we undertook protein quantification using the sensitive and accurate (i.e., dual antibodies per target) proximity extension assay (Olink Mouse

Exploratory Panel) which provided data for 92 different protein biomarkers (repeat-sample and dilution controls presented, Supplementary Fig. 11; full protein names shown in Supplementary Table 1). Since *Salmonella* Typhimurium induces a Th1 profile in the ileum[33] we first confirmed, using this analytical approach, that associated cytokines/chemokines were enhanced in ileal tissue from mice that had been challenged with *ΔaroA-Salmonella*. Of Th1 relevance, the proximity

**Fig. 4 | fgTiO₂ selectively targets PD-L1+ LysoMac and LysoDC cells. a–h** In situ, single-cell analysis of key immune-cell subtypes of the mouse subepithelial dome. **a, b** Example images immunofluorescently labelled for B220, CD3 and CD11c (i.e., identifying B-lymphocytes, T-lymphocytes and phagocytic mononuclear cells, respectively) from wild-type (WT) mice fed (**a**) without (WT) or (**b**) with (WT + TiO₂) fgTiO₂ dietary supplementation (0.0625% w/w of diet). fgTiO₂ was measured from thresholded reflectance images. Translucent red circle-markers are placed on reflectant foci to aid visualisation. **c, d** Flow cytometry-type plots showing cell area/cell aspect ratio, (**e–g**) immunofluorescence distributions and (**h**) cell counts for B220 + , CD11c+ and CD3+ cells ($n = 3$ animals). All measures remained similar regardless of fgTiO₂ feeding (cell count comparison (**h**), $P = \geq 0.26$, two-sided unpaired samples T-tests). **i–l** fgTiO₂-recipient cells of the subepithelial dome were CD11c+ in - 95% of cases. **m–o** These cells also exhibited a unique autofluorescent signature as previously described for subepithelial dome LysoMac and LysoDC cells. **o** Again, this was true for almost all fgTiO₂ + /CD11c+ cells and, equally, these

autofluorescent cells were similarly present in (**n**) controls without fgTiO₂ feeding (further data shown, Supplementary Fig. 8). **p–u** Montaged CD11c + /fgTiO₂+ or CD11c + /fgTiO₂- cell populations from the SEDs of mice fed the fgTiO₂ supplemented diet visualised in terms of (**r, s**) autofluorescence signature. fgTiO₂ is seen to both selectively and specifically target the autofluorescent cells and (**t, u**) the diversity of CD4 and MHCII expression confirms that both LysoMac and LysoDCs take up fgTiO₂. **v–z** fgTiO₂+ cells of the SED expressed the immunotolerance marker programmed death ligand 1 (PD-L1) whereas fgTiO₂+ pigment cells at the base of the Peyer's patch tended to lose PD-L1 expression (**y** versus (**z**), respectively). **w** PD-L1 expression in the Peyer's patches was not significantly perturbed by feeding the fgTiO₂-supplemented diet ($P = 0.34$, $z = -0.96$, two-sided Wilcoxon rank-sum analysis, $n = 6$ animals). The image-data presented in (**i, m, p–u, x–z**) were collected from SED tissue sections from $n = 3$ mice. Scale bars: **a, b** = 50 μm; **i, m, p–u** = 10 μm; **x** = 500 μm; **y, z** = 10 μm.

extension panel detects IL-1 (α&β), TNFα, IL-6, CXCL9 and CCL3[33,34] and four of these markers (CCL3, CXCL9, IL1β and TNFα) showed significantly ($P \leq 0.05$, two-sided Wilcoxon rank-sum tests) increased protein expression in the ileum of *ΔaroA-Salmonella*-exposed mice versus unexposed matched controls (Fig. 5e). Next we considered the effect on the ileum of chronic oral exposure to fgTiO₂ (i.e., mimicking the Peyer's patch loading seen in humans) in the absence of *Salmonella*. The proximity extension assay data showed remarkable consistency between the two groups for all measurable proteins and, therefore, no suggestion that long-term fgTiO₂ ingestion initiates inflammatory signalling in the ileum of wild-type mice (Fig. 5f and Supplementary Fig. 12). In a similar fashion, we considered the effect on the ileum of chronic oral exposure to fgTiO₂ plus *Salmonella* infection. Again, not one protein differed significantly in its expression between the groups receiving *Salmonella* and *Salmonella* + fgTiO₂ (Fig. 5f and Supplementary Fig. 12) indicating no measurable augmentation of fgTiO₂ on *Salmonella*-induced inflammatory signalling in the ileum of wild type mice, in spite of substantial same-cell uptake of the bacteria and the fgTiO₂ particles. To ensure that some small effect, localised to the Peyer's patch as the site of particle accumulation, had not been missed (e.g., diluted-out by surrounding, unaffected tissue) we made the same analyses for a potential augmentation effect using ileal tissue samples that were trimmed tightly to either side of Peyer's patch such that each sample primarily consisted of only the patch itself. But, again, no significant differences in protein expression were measurable by the Olink array (Fig. 5g and Supplementary Fig. 13).

### Cellular dosimetry and ileal tissue distribution

Finally, we considered what factors influenced fgTiO₂ distribution in the Peyer's patch. In the absence of *Salmonella* challenge, we observed fgTiO₂ in autofluorescent cells of the Peyer's patch SED region, as well as some further towards the base of the patch, at both the +3 and +28 day time-points (Fig. 6a/b), consistent with observations in Mouse Study 1. Quantitatively, however, between the two time points, there was a measurable increase in the number of fgTiO₂-loaded vesicles per cell, without a change in load per vesicle, in the mid-region of the follicle (Fig. 6c–j), suggesting a natural history of vesicular inheritance by certain cells and gradual development towards the well-reported basal pigment cells[15,16,20]. Along this pathway, cells maintained the same autofluorescent profile and mixed MHCII and CD4 expression as described in Mouse Study 1 suggesting that any such process does not give rise to a new population of cells. Visually however, with increasing distance from the SED, an increase in CD4 expression in conjunction with a decrease in MHCII expression was observed, suggesting a shift to sequestration in cells with the longer-lived, LysoMac phenotype (shown, Supplementary Fig. 14).

A more remarkable influence on the spatial distribution of fgTiO₂ was caused by the single oral dose of *ΔaroA-Salmonella*. At +28 days, SEDs were almost entirely devoid of autofluorescence-positive cells or

fgTiO₂ (Fig. 6k) and signal appeared deeper in the Peyer's patch— notably the interfollicular region (Fig. 7a, b). Consistently, it has been reported that oral danger signals, including *Salmonella*, induce migration of prior-absorbed plastic microparticles from the Peyer's patch SED to the interfollicular region[35]. In addition, some cells of the regular villous lamina propria were now positive for fgTiO₂. Many of these cells showed an autofluorescent signature and per cell loading with particles was low (Fig. 7c–e) and, yet, LysoDCs have not been reported in the villous lamina propria so the mechanisms underlying this observation are currently unclear.

## Discussion

In this work, we have exploited the growing accessibility of machine learning methods[36] to enable precise analyses of complex images yielding in situ single-cell and single-vesicle data from tissue samples with full locational data retained. Particle selectivity of cellular targets in distinct tissue regions can thus be assessed[37,38]. Access to quantitative single-cell and vesicle measurements provided an opportunity to compare and establish cell-dose equivalence between humans and mice. The findings move the field forward in four important areas of persistent particle exposure via the oral route, namely: (i) human relevance (ii) mechanism of absorption and intestinal distribution (iii) role in intestinal inflammation and (iv) provision of a model for cogent risk assessment.

### Human relevance

The human small intestine is the only known site of cellular accumulation of dietary microparticles, as exemplified by fgTiO₂[16,18,20] (Fig. 1). We show here that all features of this can be qualitatively and quantitatively recapitulated in a murine model, but fully achieving this requires exposure to human-relevant microbial danger signals. Consistently, in this work, murine dosing with fgTiO₂ was undertaken in a manner that reflects real-life exposures (i.e., via the diet at relatively low levels for a long period of time). Our findings contrast with a prior study, which used bolus dosing via oral gavage or surgical intervention in fasted animals, where direct fgTiO₂ uptake via the epithelium was reported[39]. Future work may wish to consider the necessity of natural ingestion and concomitant normal microbial exposures, versus non-physiological dosing in specific pathogen free environments, to recapitulate the human situation and address risk accurately for oral particle exposures.

### Mechanism of absorption and intestinal distribution

Seminal early work from Frey and Mantis[40,41] showed that the mucus-rich glycocalyx markedly impairs model particle uptake by the regular villous epithelium, whereas the near-absence of these features overlying Peyer's patches allows direct access of particulates to M-cells. These cells have an extraordinary ability to engulf luminal particles and macromolecules, passing them through to abutting cells at their basolateral membrane. We first confirmed, unequivocally, that this uptake mechanism holds true for fgTiO₂ and, next, demonstrated that

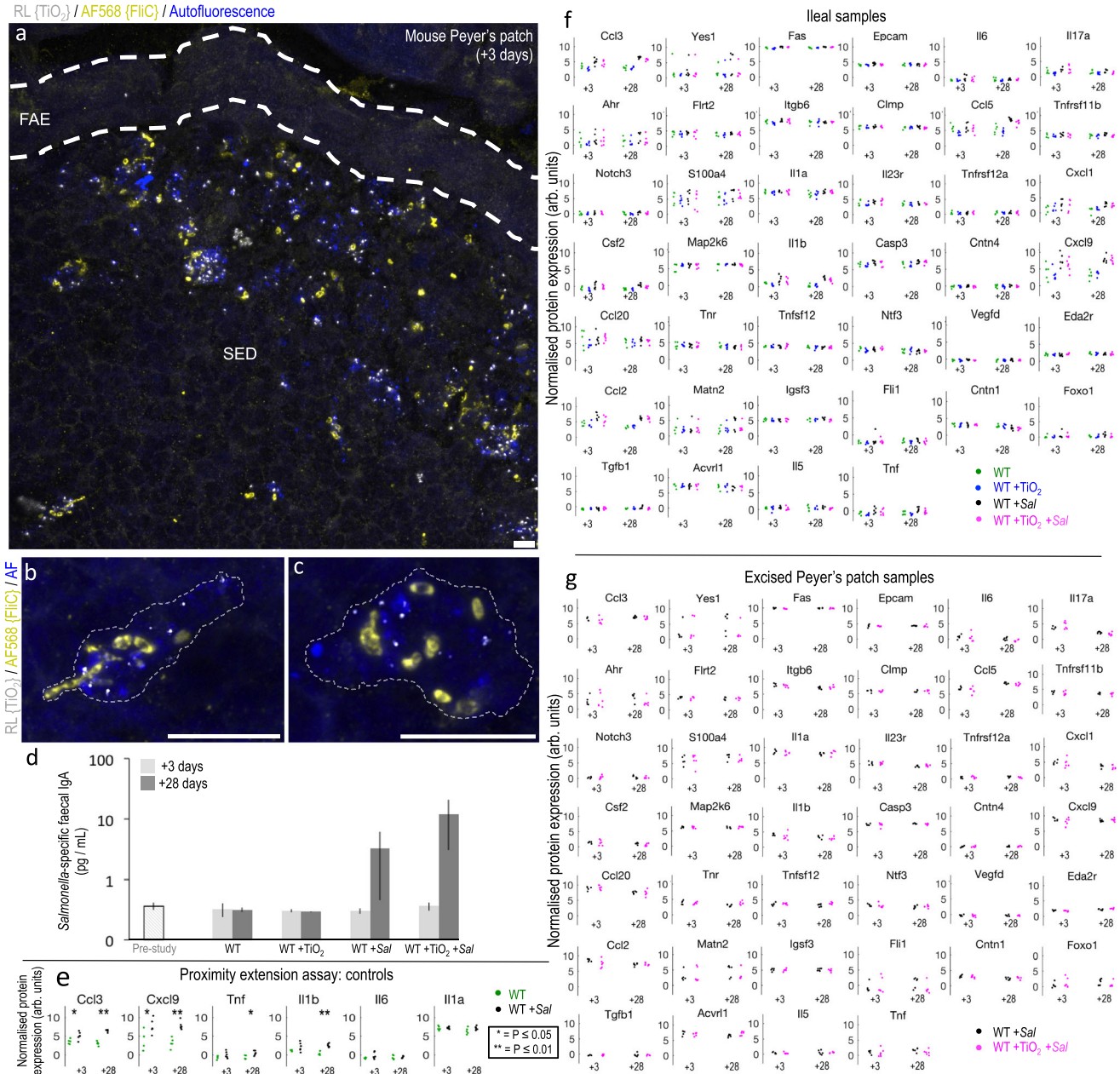

**Fig. 5 | fgTiO₂-*Salmonella* interactions.** Mouse Study 2 used a 16-week fgTiO₂ feeding period before switching all animals to a normal diet (i.e., without fgTiO₂-supplementation) then orally inoculating half with attenuated, *ΔaroA-Salmonella*. Tissues were harvested +3 or +28 days after infection. **a–c** Z-stack maximum projections showing immunofluorescent labelling for flagellin (FliC) to detect the *ΔaroA-Salmonella* in the Peyer's patch SED region. Marked accumulation of both fgTiO₂ and *ΔaroA-Salmonella* was observed in autofluorescent, phagocytic mononuclear cells at the +3 timepoint. **d** Faecal ELISA studies confirmed a *Salmonella*-specific IgA response at the +28 timepoint (bars represent medians with interquartile range error-bars, *n* = 3–6 samples per treatment group). **e–g** Protein expression analyses of ileal tissue digests by proximity extension assay (OLINK mouse exploratory panel). **e** Comparison of ileal tissues taken from wild-type (WT) mice unexposed to fgTiO₂ but with (WT +*Sal*) or without (WT) *ΔaroA-Salmonella* infection confirmed significant increases key cytokines/chemokines associated with a Th1 immune response (*P* ≤ 0.05, two-sided Wilcoxon rank-sum test, *n* = 5–6

tissues per group). **f** Inflammation-associated protein expression analyses of ileal tissue digests. At each timepoint, no significant differences in protein expression were observed between the WT and WT +TiO₂ (*P* ≥ 0.53, two-sided Wilcoxon rank-sum test, *n* = 3 animals per group) or WT +*Sal* and WT +TiO₂ +*Sal* groups (*P* ≥ 0.10, two-sided Wilcoxon rank-sum test, *n* = 6 animals per group). Full dataset is shown in Supplementary Fig. 12. **g** Peyer's patch-focussed inflammation-associated protein expression analyses using carefully-excised patch digests in mice treated with either *ΔaroA-Salmonella* alone (WT +*Sal*) or with fgTiO₂ and *ΔaroA-Salmonella* (WT +TiO₂ +*Sal*) at the +3 or +28 timepoints. At each timepoint, no significant differences were observed between the two groups (*P* ≥ 0.20, two-sided Wilcoxon rank-sum test, *n* = 6 animals per group) demonstrating no interaction of fgTiO₂ and *ΔaroA-Salmonella* despite (**a–c**) heavy loading into the same cells (full dataset shown in Supplementary Fig. 13). OLINK protein abbreviations are defined in Supplementary Table 1. All *p*-values are available for download at the BioStudies database under accession number S-BSST875. Scale bars: **a–c** = 10 μm.

specific SED immuno-competent phagocytes (LysoMacs and LysoDCs) are the direct recipient cells that accumulate cargo from the M-cell. Taking all of our data together, we believe that there is an on-going process of slow re-arrangement, typified by vesicular inheritance by

some of the cells and gradual particle-loaded-cell movement to the base of the patch. This migration, which must ultimately result in mature pigment cell formation[16,18], appears to be accelerated by oral exposure to the bacterial pathogen *S.* Typhimurium which,

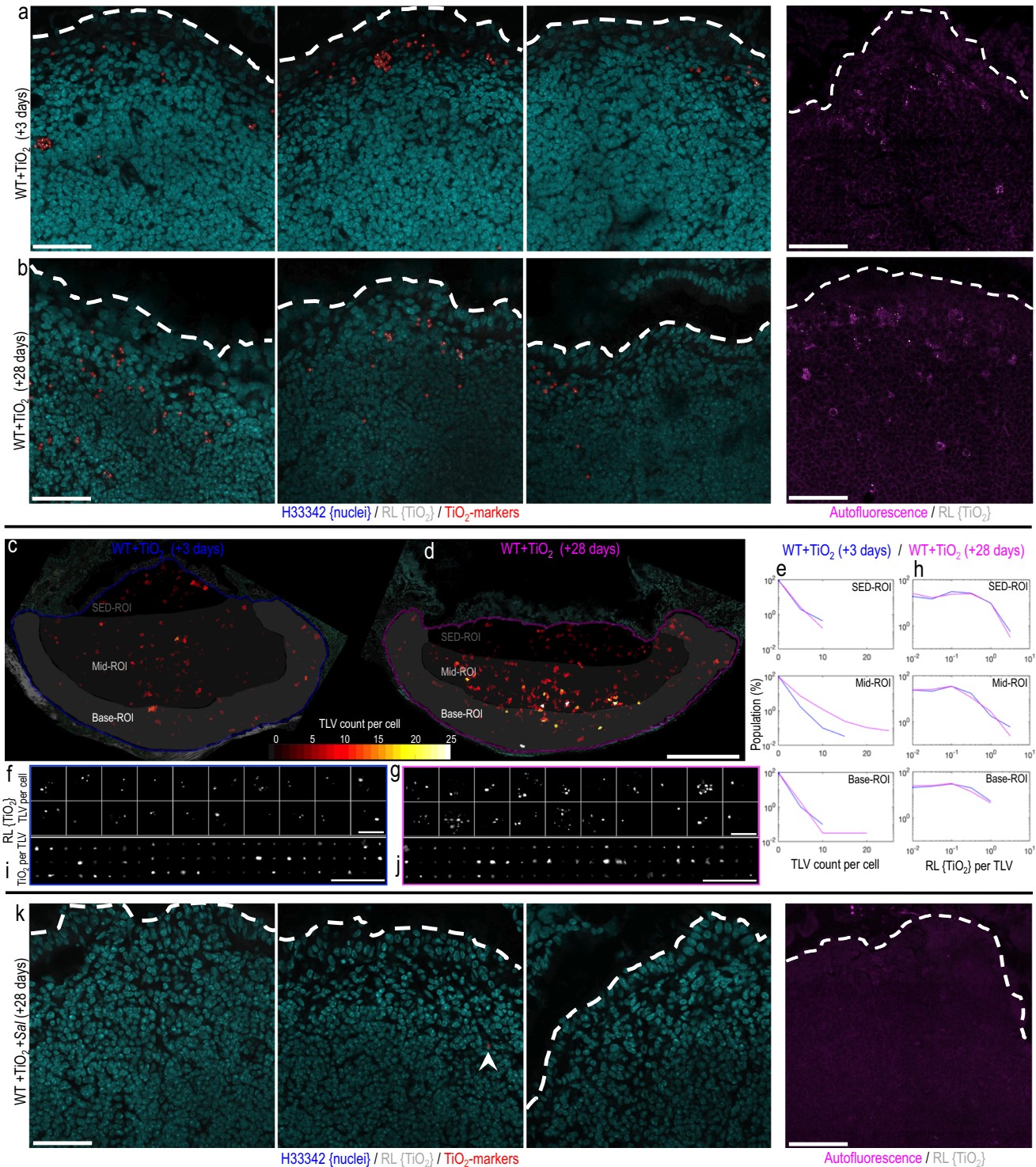

additionally, directs fgTiO$_2$ to the interfollicular region of the patch—consistent with apical stimulation of LysoDCs that then move to naïve T cells[42,43]. In addition, chasing fgTiO$_2$-exposed animals with *S.* Typhimurium led to the appearance of some particles in cells of the villous lamina propria which, without such infection, yields no particulate signal. Again, building on formative work on the fate of reporter particles of fluorescent plastic in the Peyer's patch[35], it is likely that the danger signal effect holds true for oral exposure to soluble enterotoxin adjuvants as well as pathogenic bacteria (i.e. multiple human-relevant danger signals). Continuous exposure to fgTiO$_2$, as happens for many human populations, would then re-load the apical (SED) phagocytes

until another danger signal event occurs. Thus, we describe the natural history of persistent particle routing through—and beyond—the Peyer's patch following oral exposure. This explains basal pigment cell accumulation of multiple particle types, including titanium dioxide, silica and aluminosilicates, in human Peyer's patches as well as the occurrence of sporadic particles elsewhere in cells of the small intestine[16,18,20].

## Role in intestinal pathology
By extension of the above observations, our findings also imply that fgTiO$_2$ is not in itself perceived as a danger signal in the intestine of

**Fig. 6 | Cellular dosimetry of fgTiO$_2$ across timepoints and *Salmonella*-induced migration. a, b, k** Confocal reflectance micrographs showing the subepithelial dome (SED) tissue region ($n = 3$ animals per group). After the 16-week fgTiO$_2$ feeding period, all animals were switched to a normal diet (without fgTiO$_2$ supplementation) and half were orally dosed with *ΔaroA-Salmonella* before tissue harvest at +3 or +28-day timepoints. **a, b** In the animals fed the fgTiO$_2$-supplemented diet without *Salmonella* exposure (WT +TiO$_2$), fgTiO$_2$-loaded cells were present in the SEDs at both +3 and +28 timepoints. **c–j** To gain insights into cellular dosimetry changes across the two timepoints, image analysis of Peyer's patch transverse sections was used to measure the number of reflectant foci per-cell (i.e., the fgTiO$_2$-loaded vesicle (TLV) count per-cell) and the amount of reflectance per foci (i.e., equivalent to the fgTiO$_2$ dose per-vesicle). **d, e** At the +28 timepoint, the TLV count per-cell was elevated with a new population of cells with TLV counts >13 forming in the mid-to-base regions of the follicle. **f, g** Randomly sampling and montaging the reflectance (RL) information in cells from each of the two timepoints

visually demonstrates this, showing (**g**) more heavily-loaded cells at the +28 timepoint. **h** Moving from cell to vesicle information, in contrast, the amount of reflected light per individual foci remained near-identical across the two timepoints and this was again borne out visually when (**i, j**) individual TLVs were randomly montaged. Collectively, the results suggest that once fgTiO$_2$-loaded cells move below the SED, there is a process of vesicular inheritance by certain cells over time, eventually yielding heavily-loaded pigment cells at the follicle base–as is well-described in humans (Fig. 1). **k** Challenge with *ΔaroA-Salmonella* (WT +TiO$_2$ +*Sal*) markedly changed this picture ($n = 4$ animals per group). At the +28 timepoint, SEDs were almost completely devoid of both fgTiO$_2$ and autofluorescence-positive cells. Out of four SEDs imaged, only one reflectant foci was detectable (indicated by arrow) showing marked migration of fgTiO$_2$ from the SED tissue compartment after *ΔaroA-Salmonella* exposure. Scale bars: **a, b, k** = 50 μm, **c, d** = 250 μm, **f, g, i, j** = 10 μm.

wild-type mice since a secondary (microbial) signal is still required for the acute migration. Moreover, using highly sensitive and selective protein analyses that captured a broad array of biomarkers of immuno-inflammatory signalling, we were unable to identify either direct or augmented perturbation of tissue homeostasis at the site of particle accumulation. Indeed, mammals have very likely evolved with oral exposure to other fine particles (e.g., geologically-derived and soot). Certainly, aluminosilicates, in abundance, and some forms of silica are reported in human Peyer's patches alongside titanium dioxide[16,18]. Perhaps, therefore, effective compensatory mechanisms generally exist, which may include constitutive cell expression of immuno-inhibitory PD-L1 (Fig. 4v–z).

Notwithstanding, the potential for fgTiO$_2$- induction or augmentation of intestinal pathology still deserves scrutiny. First, in subjects with Crohn's disease, fine-particle-targeted macrophages of the Peyer's patch do not express PD-L1[25]. Interestingly, Crohn's disease is a relatively modern inflammatory disorder and there is good evidence that it may initiate in the Peyer's patches[44]. The most commonly-associated single gene mutation with the disease is NOD2 and there is in vitro evidence that when NOD2 is perturbed, fgTiO$_2$ augments enhanced inflammatory activity with microbial fragments[1]. So, it should now be carefully considered what impact the ingestion of fgTiO$_2$ has on at-risk genotypes and, especially, where NOD2 functionality is diminished or removed. Secondly, persistent particulates in the size range of fgTiO$_2$ are recognised as potential immuno-adjuvants, notably in skewing the magnitude and direction of T cell proliferative responses[45]. Studies reported herein, demonstrating no measurable augmentation of intestinal inflammatory responses by fgTiO$_2$, do not preclude a separate immuno-adjuvant role for fgTiO$_2$. Ex vivo studies, using fgTiO$_2$- and/or antigen-loaded LysoMacs and LysoDCs to stimulate T cell proliferation and activation should be considered. In vivo, oral adjuvant effects of fgTiO$_2$, with a co-ingested neo-antigen, could similarly be investigated as, for example, was previously reported for lactoferrin with delayed type hypersensitivity as a read out[46]. Thirdly, it is important to note that conclusions from this work refer to the distal small intestine where there is uptake of fgTiO$_2$. As noted previously[12], however, it does not shed light on potential effects of fgTiO$_2$ where it is not absorbed such as in the large intestine. Direct luminal effects (on microbiome or apical enterocytes for example) cannot be precluded, especially given that this is where duration of exposure will be by far the greatest. Indeed, both Urrutia-Ortega et al.[47] and Bettini et al.[48] have provided evidence for direct, pro-tumorigenic effects of fgTiO$_2$ on the colon which deserves further attention[12].

**A model for cogent risk assessment**
Finally, we note that the particulate-accumulating population of apical SED cells is clearly (i) immunocompetent (i.e., non-quiescent) based upon their migration characteristics, expression of cell surface

immunoregulatory PD-L1 and their previously-described qualities in terms of antimicrobial defence[17,27,49,50] and (ii) capable of becoming heavily loaded with fgTiO$_2$. Indeed, the exquisite particle selectivity for this tissue region means that the concentration gradient for area-corrected reflectance signal was ~14-fold higher than for the Peyer's patch overall and, therefore, ~1245 fold higher than that of villous mucosa (Supplementary Fig. 15). Given that the particulate-loaded phagocytic cells make up only a minority of the SED[17,23], then the per-cell concentration of fgTiO$_2$ is very high (Supplementary Movie 1).

We know of no other active particle-funnelling and specific cell-targeting strategy elsewhere in mammalian biology so it is very likely that SED LysoMacs and LysoDCs provide the only target to enable highly sensitive, single-cell risk assessment of fgTiO$_2$ and other persistent oral particulates in vivo. By exemplification, we addressed the question of whether fgTiO$_2$ could initiate or augment inflammatory signaling in the ileum / Peyer's patches of wild type mice: a long standing question in the field. The clear answer was no, at least in wild-type genotypes, despite some in vitro evidence to the contrary[1]. To explain such differences it is worth noting that cell culture studies fail to replicate in vivo fidelity in many ways including rates of particle uptake, presence of functional lymphatics and vasculature, cellular cross-talk, compensatory mechanisms and cell migration prior to particle saturation (cell gorging). Whatever the reasons, after decades of uncertainty, other questions regarding the safety of persistent oral particles can now be precisely addressed using in vivo studies with human-relevant dosimetry and identified target cells. Indeed, understanding the consequences of LysoMac/LysoDC accumulation of persistent particles in terms of genotoxicity, adjuvanticity and any differential responses resultant from distinct host genotypes is now key to understanding human risk following oral exposure to nanoparticles and microparticles.

## Methods
### Ethical approval of animal and human studies
The mouse and human tissue studies described in this work complied with all relevant ethical regulations. The first mouse fgTiO$_2$ feeding study (i.e., Mouse Study 1) was approved by the Grasslands Animal Ethics Committee (Palmerston North, New Zealand) in accordance with the New Zealand Animal Welfare Act 1999. The second mouse fgTiO$_2$ feeding study (i.e., Mouse Study 2) was performed at Cambridge under the authority of Home Office personal license number PIL IEDB62633, and project license number PPL PF86EABB1. Following written informed consent, and after approval from the Regional Ethical Review Board, Linköping, Sweden, human surgical specimens containing Peyer's patches were collected from the neo-terminal ileum from patients with Crohn's disease or colonic cancer. Studies using human tissues at Cambridge were also approved by the UK NHS Health Research Authority, North West–Greater Manchester East Research Ethics Committee, REC reference 18/NW/0690.

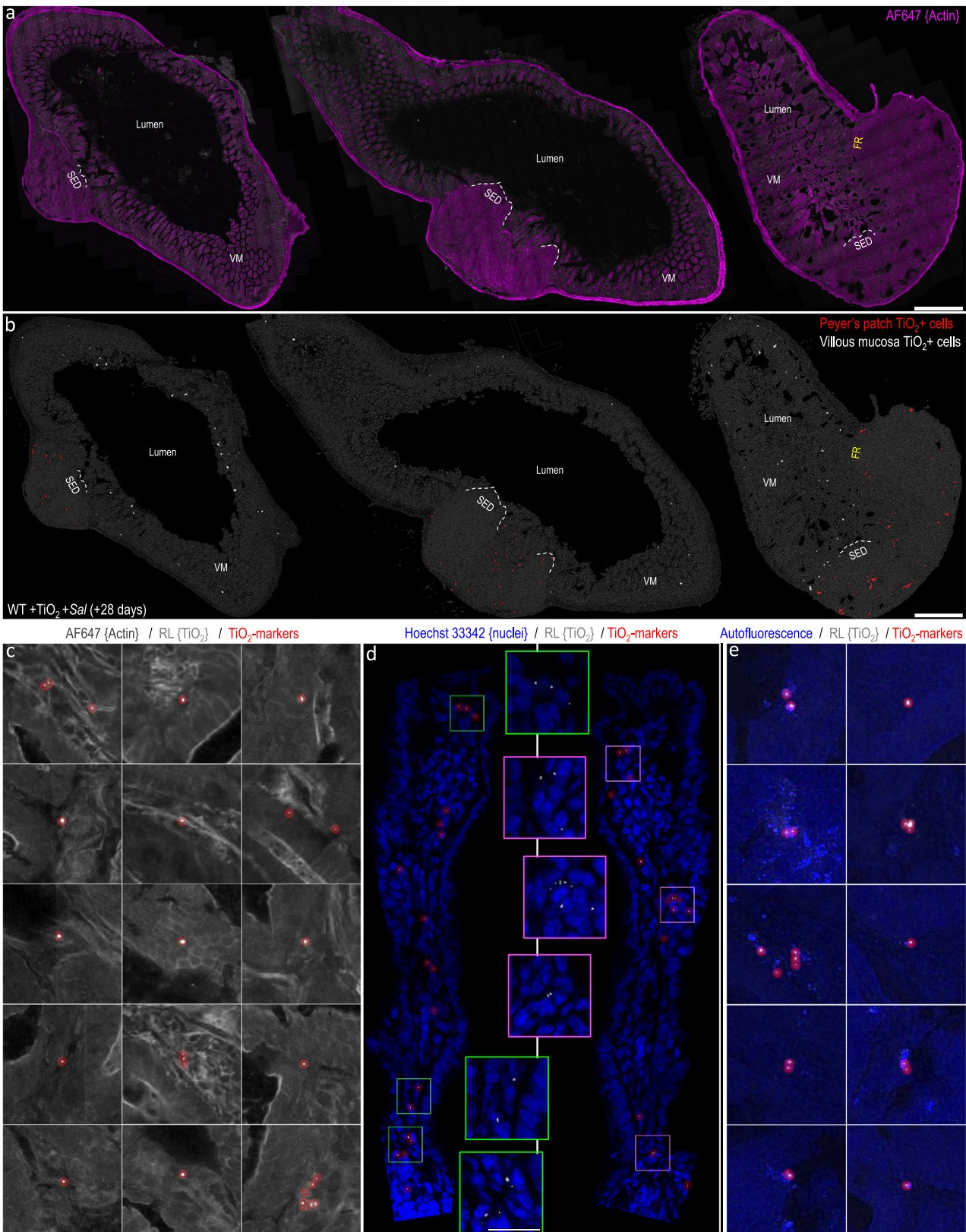

## Mouse study 1

Food-grade, anatase TiO$_2$ particles (fgTiO$_2$) were purchased from Sensient Colors (St. Louis, MO, USA). Physico-chemical characterisation of these particles showed a particle diameter distribution of ~ 50–300 nm, with a median diameter of ~ 130 nm[1]. As described and characterised previously[15], to enable successful oral delivery, the standard rodent diet American Institute of Nutrition (AIN)−76A was supplemented with 0.000625%, 0.00625% or 0.0625% (w/w) fgTiO$_2$ (translating to ~1, 10 or 100 mg fgTiO$_2$/kg body-weight/day, respectively[15]). These diets were prepared by Research Diets (New Brunswick, NJ, USA), with straight AIN-76A without fgTiO$_2$ supplementation used as the negative control diet. Forty-eight, six-week-old

**Fig. 7 | Ileal distribution of fgTiO$_2$ following *Salmonella* challenge. a, b** Single-cell image analysis of tilescanned confocal reflectance images collected from transverse sections of mouse ileal tissue from the fgTiO$_2$-exposed, *ΔaroA-Salmonella*-infected treatment group (+28-day timepoint, $n = 3$ animals; FR denotes follicular region with no follicle-associated epithelium). **b** As in Fig. 6, the sub-epithelial dome (SED) regions were largely devoid of fgTiO$_2$ with the majority of the positive cells located at the follicle base in the pigment cell zone or in the inter-follicular regions (fgTiO$_2$-positive cells in the Peyer's patches are displayed in red). **b** In contrast to all previous data collected without *ΔaroA-Salmonella* exposure, fgTiO$_2$-positive cells were now also present in the villous mucosa (VM) (displayed in white). **c** Montaging positive events from villous mucosa regions showed that particle loading was low compared to Peyer's patch cells with the majority containing single reflectant foci. **d** High resolution imaging of optically cross-sectioned villi confirmed reflectant foci deep in the lamina propria in the same manner as observed in humans (Fig. 1c). **e** The majority of the fgTiO$_2$-positive events in the villous mucosa also displayed the same autofluorescence signature as was observed in the particle-recipient cells of the SED. The image-data presented in **c-e** were collected from ileal sections from $n = 3$ mice. Scale bars: **a, b** = 250 μm; **c–e** = 20 μm.

mice (*Mus musculus* C57BL/6 J, 50:50 male:female) were sourced from the AgResearch Ruakura Small Animal Colony and randomly split into four feeding groups (0.00%, 0.000625%, 0.00625% or 0.0625% (w/w) fgTiO$_2$/AIN-76A, $n = 6$ male and $n = 6$ female per group) with normal feed intake and weight-gain confirmed bi-weekly[15]. Mice were housed under a standard 12 h light, 12 h dark cycle, with a temperature maintained between 20 and 24 °C and a relative humidity of 40–60%. After eighteen weeks feeding, mice were euthanised by CO$_2$ asphyxiation and cervical dislocation. The gastrointestinal tract was quickly removed and placed in cold phosphate-buffered saline (PBS). Ileal tissues containing the most distal Peyer's patches (i.e., closest to the ileal/caecal junction) and adjacent caecal patches were excised and frozen on a cold, stainless steel bar embedded in dry ice. Tissues were then transferred into cryomolds filled with pre-cooled optimal cutting temperature compound (OCT) and transferred to −80 °C for storage. In previous work using tissues from the same animals, we established that fgTiO$_2$ intake and delivery into Peyer's patches was not significantly different between male and female animals[15]. For these reasons and to maximise statistical power, data across males and females were combined and anlaysed together in the presented work. During analysis, sex information for individual tissue sections was not retained so the presented data cannot be disaggregated by sex.

## Mouse study 2

Female mice (*Mus musculus* C57BL/6NCrl) were sourced from Charles River Laboratories (UK). They were housed under a standard 12 h light, 12 h dark cycle, with a temperature maintained between 20 and 24 °C, a relative humidity of 40–60% and were monitored for signs of normal health, food ingestion and body-weight gain throughout the study period. At the start of the study, thirty-six mice (six weeks old) were randomly divided into two feeding groups. As before, one group received the standard rodent diet American Institute of Nutrition (AIN) −76A supplemented with 0.0625% (w/w) fgTiO$_2$ (translating to approximately 100 mg fgTiO$_2$ / kg body-weight/day respectively[15]), whereas the other received straight AIN-76A without fgTiO$_2$ supplementation (i.e., the negative control diet). After sixteen weeks of feeding, all mice were switched to the negative control diet, and twelve mice from each diet-group were orally challenged with *ΔaroA*-deficient, *S.* Typhimurium strain SL3261[51] (*ΔaroA-Salmonella*). We marginally separated the administrations of bacteria and fgTiO$_2$ to (i) prevent artefactual adsorption due to co-administration (ii) enable luminal interactions between gavaged bacteria and residual luminal particles (demonstrated by faecal ICP-MS studies, Supplementary Fig. 16) should such interactions occur and (iii) ensure that some food also was in the lumen under these circumstances as would be expected with normal feeding. Mice were lightly anaesthetised with isofluorane then orally gavaged with 0.2 mL of inoculum prepared at $5.25 \times 10^9$ CFU/mL. Mice were then euthanised by CO$_2$ asphyxiation and cervical dislocation at two time-points, either +3 or +28 days after *Salmonella* inoculation (i.e., 16 weeks +3 days or 16 weeks +28 days). In this way, four treatment groups ($n = 6$ control diet, n = 12 control diet + *Salmonella*, $n = 6$ fgTiO$_2$ diet, $n = 12$ fgTiO$_2$ diet + *Salmonella*) were established, with half of the animals analysed at each of the two time-points. Alongside, faecal samples were collected at three time-points

(pre-study, 16 weeks +3 days and 16 weeks +28 days) to enable enzyme-linked immunosorbent assay (ELISA) studies for *Salmonella*-specific IgA and faecal Ti measurements by ICP-MS.

## Human tissue collection

Surgical specimens containing Peyer's patches were collected from the neo-terminal ileum from six patients with Crohn's disease or colonic cancer (3 males, 3 females; self-reported). These tissue samples were made available for microscopical analysis with group-summary information only (i.e., resection date-range 2017–2018; median age 48 years; age-range 21–77 years) so disaggregation of individual microscopy data in terms of sex was not possible. In all instances the ileal tissue specimens formed the resection margins and were macroscopically normal. Peyer's patches were confirmed through microscopical identification[52]. The tissue specimens were fixed with 4% PBS-buffered (pH 7.4) paraformaldehyde for 12 h at 4 °C before immersion in 30% sucrose and freezing in OCT (LAMB/OCT, Thermo Scientific) in cryomolds. For microscopy analyses, the OCT-embedded tissue samples were rapidly shipped overnight to Cambridge on dry ice. Alongside, three anonymised, formalin-fixed paraffin embedded (FFPE) human tissue samples containing ileal Peyer's patch lymphoid tissue were also analysed to confirm if fgTiO$_2$ could be detected in the FFPE specimen-type. Beyond tissue-type, no information about these human samples was available.

## Tissue sectioning and immunofluorescence labeling

Frozen sections were cut at 20-micron thickness using a Leica CM 1900 cryostat, picked up on Superfrost Plus slides (ThermoFisher, J1800AMNT) and rested at room temperature for 2 h prior to labelling. FFPE sections were cut at 5-micron thickness, then dewaxed and rehydrated by baking at 60 °C for 1 h, transferring through two changes of xylene, a reverse ethanol series (100%, 70%, 50%, 10%) followed by 1 minute rehydration in water. Sections were ringed with hydrophobic barrier pen (Vector Laboratories, H-4000) and cryostat sections fixed by exposure to fresh 4%, 0.1 M PBS-buffered paraformaldehyde (pH 7.4) for 10 min. All subsequent steps were conducted under gentle agitation on a rotating shaker. To facilitate antibody penetration, sections were permeabilised for 90 min using 0.3 % (vol/vol) Triton X-100 in 0.1 M PBS (pH 7.4). Block buffer (10% goat serum, 2% bovine serum albumin, 20 mM glycine diluted in 25 mM Tris-buffered (pH 7.4) saline (TBS)) was then added to all sections for 1 h. Primary antibodies or concentration-matched isotype controls were diluted in block-buffer and incubated at 4 °C overnight. All subsequent steps took place at room temperature. Sections were washed with three changes of block-buffer. Secondary antibodies (where directly-conjugated primary antibodies were not used) were diluted 1:400 in block buffer and incubated for 4 h. All antibody concentrations, fluorophore conjugations and manufacturer information are provided in Supplementary Table 2. 500 nM phalloidin-AlexaFluor 647 (ThermoFisher, A22287) and 2 μg/mL Hoechst 33342 (Thermo-Fisher, H3570) were also included alongside secondary antibodies to simultaneously label cell outlines (i.e., cytoskeletal f-actin) and cell nuclei (respectively). Sections were then washed once with TBS prior to mounting with #1.5 coverslips in Prolong Diamond (ThermoFisher, P36965).

## Confocal microscopy & imaging controls

Fluorescence and reflectance image-data (2048×2048 pixels per tile) were collected using a diode-controlled (retrofitted) Zeiss LSM780 confocal microscope equipped with plan-apochromat 63X/1.4 and 40X/1.3 oil-immersion objectives. Fluorescence and reflectance data were collected at the same time during the same imaging run by sequential scanning with reflected (i.e., back-scattered) light from a 488 nm excitation laser collected in the range 485–491 nm to detect $fgTiO_2$[15]. Where tilescanning was used, images were acquired with 10% edge overlap. Tiles were stitched using the Zen Black 2011 SP2 software (Carl-Zeiss, UK) using the nuclei and actin channels for registration and a strict correlation threshold setting of 1.0. For all datasets under comparison, images were obtained in a single run using identical settings. For the reflectance imaging, tissues from negative control mice (i.e., fed the AIN-76 diet only, so no $fgTiO_2$ present) were used to determine the image acquisition settings and thresholding for $fgTiO_2$ detection (strategy shown, Supplementary Fig. 1). For immunofluorescence work, supporting secondary-only and concentration-matched isotype antibody controls in tissue-matched serial sections were included. For datasets collected as sets of single images from the villous mucosa (VM), sub-epithelial dome (SED), germinal center (GC) and TCZ (T-cell zone) regions of mouse Peyer's patches, landmarks (e.g., villi / follicle-associated epithelium) and T cell (CD3), B cell (B220) and mononuclear phagocyte (CD11c) immunofluorescence staining in adjacent sections were used to inform imaging locations. To define the autofluorescence signature of the $fgTiO_2$-recipient cells in the mouse Peyer's patches, lambda scans were performed using the LSM 780 microscope. Using unstained tissue sections, autofluorescence upon excitation from a 405 nm laser was detected in ~10 nm intervals in the range 415–700 nm (spectra presented, Supplementary Fig. 8).

## Deconvolution

Z-stacked image-data (pixel size (XYZ) 1024x1024x123; voxel size (XYZ) 26x26x100 nm) of $fgTiO_2$-recipient cells was collected using a Zeiss LSM 780 confocal microscope equipped with a 63X/1.4 plan-apochromat oil immersion objective using murine tissue sections embedded in Prolong Glass (P36984, ThermoFisher) anti-fade reagent (refractive index ~ 1.5). Cell autofluorescence was excited using the 405 nm laser. The image-stacks were registered to correct spatial drift using StackReg[53]. The Born and Wolf scalar-based diffraction model[54] was then used to estimate a theoretical point spread function assuming a refractive index of 1.51 and a autofluorescence emission maximum of 495 nm (measured, Supplementary Fig. 8). The registered Z-stack was then deconvolved using the Richardson-Lucy total variation algorithm (75 iterations, regularisation 1e-12) within the open-source DeconvolutionLab2[55] software.

## $fgTiO_2$ detection using reflected light and circle-marker placement

To enable $fgTiO_2$ detection in biological tissue sections, reflected light and cell structure (i.e., nuclei / actin) fluorescence information were collected at the same time using a Zeiss LSM780 confocal microscope. The reflectance images were thresholded to isolate the punctate, reflectant foci caused by the $fgTiO_2$ particles from the background light scatter caused by the biological tissue. As the background level was similar in both mouse and human specimens (shown, Supplementary Fig. 4) the same threshold was used for both mouse and human images, and was determined as the level required to remove all signal in the reflectance channel for images collected for tissue sections taken from the negative control mice (i.e., fed the AIN-76 diet only, so no $fgTiO_2$ present). This approach was then further validated using correlative scanning electron microscopy and energy dispersive X-ray analyses to confirm that the signal remaining after thresholding contained the expected X-ray signal for titanium (strategy shown,

Supplementary Fig. 1). Given the small size of the reflectant foci caused by $fgTiO_2$, a translucent circle-marker was placed on each pixel above the threshold using the insertShape function in MATLAB R2021a (MathWorks) to aid visualisation given the relatively small size of display in the Figure panels[15]. Examples of the raw signal and subsequent marker placements are shown for mouse and humans in Supplementary Fig. 5. Example data and code demonstrating thresholding and circle marker placement are available at the BioStudies database under accession number S-BSST875.

## Correlative confocal-electron microscopy and energy dispersive X-ray analyses

Once $fgTiO_2$ detection by confocal reflectance microscopy was completed, Prolong-mounted coverslips were removed from the tissue sections by overnight incubation in PBS at 37 °C. Slides were air dried at room temperature for 24 h, cut down and mounted on aluminum SEM stubs using colloidal graphite and coated with a thin (~15 nm) layer of amorphous carbon (evaporative coater). Tissue sections were then loaded into a Helios G4 CX Dual Beam high-resolution, monochromated, field emission gun, scanning electron microscope (SEM) equipped with a precise focused ion beam (FIB) and in-lens secondary electron and circular backscattered detectors (FEI/ThermoFisher). Secondary and backscattered electron images were collected at 3–5 kV as tilescans for the complete tissue surface, permitting manual overlay onto the previously-collected confocal tilescans. To enable correlative confocal-SEM imaging, the outline of the Peyer's patch–particularly the follicle associated epithelium and basal muscularis layer–were manually registered together using image overlay, enabling triangulation to target specific regions containing reflectance-positive cells. The correlative confocal-to-EDX analyses of the human tissues were then carried out directly in the SEM, using a 150 mm² silicon drift detector and AZTEC software (Oxford Instruments). For the correlative scanning transmission electron microscopy (STEM) used to confirm single-particle detection by reflectance microscopy, thin lamella of the tissue sample were prepared in the SEM via the in situ lift-out method. Once sites-of-interest had been identified, 500 nm of electron beam platinum (Pt) was deposited (at 5 kV, 6.4 nA for the electron source) to the surface of the target area. This was followed by a second Pt layer (1 µm) using the FIB (at 30 kV, 80 pA for the liquid gallium (Ga) ion source). A bulk lamella was initially cut using the FIB (at 30 kV, 9 nA), before the final cut-out was performed (at 30 kV, 80 pA). The lamella was attached, using ion beam Pt, to a copper FIB lift-out grid (Omniprobe, USA) mounted within the SEM chamber (i.e., in situ). Final thinning and polishing of the lamella to electron translucency was performed with a gentle polish/clean using the ion beam (5 kV, 41 pA). A lamella was then transferred into a Titan3 Themis G2 300 kV S/TEM equipped with an S-TWIN objective lens, monochromator, HAADF detector and a Super-X 4-detector EDX system in a double-tilt specimen holder (FEI/ThermoFisher). Transmission electron microscopy (TEM) images and on-zone diffraction patterns were recorded using a OneView CMOS camera (Gatan) and STEM-EDX elemental maps were recorded and processed using the Velox software (FEI/ThermoFisher).

## Cell segmentation of intestinal tissue images

In previous work, we carefully developed and validated staining and image analysis methods enabling extraction of single-cell data from confocal microscopy images of gastrointestinal tissues[37]. Based on this framework, three cell segmentation pipelines were used to accommodate different tissue types and differing availabilities of training data. To confirm the reliability of our previously-published pipelines with the image-data presented here, the accuracy of the outputs from all three cell segmentation pipelines was assessed relative to manually-segmented cells by Jaccard index (presented, Supplementary Fig. 2).

## Cell segmentation: mouse villous mucosa images

Villous image-fields were directly segmented into cell-objects using the freely-available CellProfiler software[56]. Using the Hoechst 33342 fluorescence information, nuclei were first segmented as primary objects. These were then used as seeds during a marker-controlled watershed process to segment each cell's outline using fluorescence information from the actin channel. The process is schematically demonstrated in Supplementary Fig. 17 with the complete CellProfiler pipeline shown in Supplementary Note 1. Test images and the CellProfiler pipeline are also available for download at the BioStudies database under accession number S-BSST875.

## Cell segmentation: mouse lymphoid tissue images

Lymphoid tissues (e.g., Peyer's patches and caecal patches) exhibit high cell densities leading to poor cell segmentation accuracies using the simple, marker-controlled watershed approach used for the villous mucosa images[37]. Pixel-classification machine learning can be used to alleviate this problem through the generation of probability images that more precisely delineate each cell's boundary[37]. Here, in mouse lymphoid tissues, these probability images were provided using a 2-D U-Net fully convolutional neural network[57] deployed in MATLAB R2021a (MathWorks, MA). The network has a 256x256x2 (x, y, channels) input layer and uses the nuclear and actin fluorescence information to predict the probability that each pixel belongs to either cell outline, intracellular environment or background classifications. The training data consisted of twelve lymphoid tissues images (each containing $2138 \times 1900$ annotated pixels covering ~16,000 cells prepared by an experienced cell biologist). The image-data flows through a four-layer contracting convolutional path before complete up-convolutional expansion, yielding probability images that exactly match the input-image dimensions. The network was trained for 50 epochs using a batch-size of 12 for 1200 iterations/epoch. Patches were shuffled every epoch and augmented by simple, random x/y reflection and rotation. Model training was optimised under stochastic gradient descent using cross-entropy loss. The initial learning rate was 0.05, dropping every 10 epochs by 0.1 under momentum 0.9 and L2 regularisation $1 \times 10^{-4}$. The resultant cell outline probability images were then loaded into the CellProfiler pipeline alongside immunofluorescence and reflectance information collected by the confocal microscope and were used to enable segmentation of individual cells via an IdentifyPrimaryObjects module. This process is schematically demonstrated in Supplementary Fig. 18. The complete CellProfiler image analysis pipeline is presented in Supplementary Note 2. All U-Net code, the trained network with test-data and the complete CellProfiler pipeline are available for download from the BioStudies database under accession number S-BSST875.

## Cell/TLV segmentation: correlative mouse-human dosimetry

For the correlative analyses investigating the relative amounts of fgTiO₂ delivered to individual SED cells in mice versus in normally-exposed humans, frozen ileal tissue sections containing Peyer's patches were collected at random from six normally-exposed humans, and six mice fed the AIN-67 diet supplemented with 0.0625% (w/w) fgTiO₂. In every instance, care was taken to collect tissue sections with clear follicle-associated epithelium (i.e., without any overlying villi) to ensure maintenance of similar histological positioning near the center of the dome of each Peyer's patch. Reflectance confocal images of the murine and human tissues were then acquired in a single run under identical settings. After acquisition, the image-fields per specimen were manually laid out in order of lowest-to-highest fgTiO₂ accumulation to visually present the spectrum of (wide) variation in fgTiO₂ cellular loading observed across samples in both the mouse and human tissue specimens. This was followed by quantitative image analysis to determine the thresholded reflectance per unit tissue area; then the integrated intensity of the threshold reflectance information per-cell and per-vesicle objects (method exemplified, Supplementary

Fig. 3). To extract the single-cell and single-vesicle data, a CellProfiler pipeline was used. To tackle cell segmentation in lymphoid tissues with high cell densities we have shown in previous work[37] that pixel-classification machine learning can be used to produce probability images that more precisely delineate each cell's boundary improving segmentation accuracies when compared to using the raw fluorescence information. Unlike in mouse lymphoid tissues (method described above), because large amounts of manually-annotated training data for the human lymphoid images were not available, these probability images were created by sparse annotation pixel-classification machine learning in the ILASTIK software[58]. Two ILASTIK pixel-classification projects were created (one for the human images, one for the mouse). Pixel annotations representing intracellular environment, cell outlines and other classifications were made on the fluorescence channel describing the cell outlines (i.e., the actin channel) (method shown schematically, Supplementary Fig. 19). The ILASTIK software then generated probability images representing the probability of each pixel belonging to each classification. The resultant cell outline probability images were loaded into the CellProfiler pipeline alongside the fluorescence and reflectance information captured by the confocal microscope and were used to segment individual cells via an IdentifyPrimaryObjects module. As part of this pipeline, the reflectant foci representing fgTiO₂ events were also directly-segmented from the thresholded reflectance images into fgTiO₂-loaded vesicle (TLV) objects using a second, IdentifyPrimaryObjects module (method schematically demonstrated, Supplementary Fig. 20). The complete image analysis pipeline is presented in Supplementary Note 3. Test images and the complete CellProfiler pipeline are available for download from the BioStudies database under accession number S-BSST875. The accuracy of the segmented-cell outputs was checked by Jaccard index relative to manually-segmented cells (presented, Supplementary Fig. 2). The accuracy of TLV segmentation from the raw reflectance information is explored in both mouse and human images in Supplementary Fig. 5.

## Cell/TLV feature extraction

Raw immunofluorescence data were pre-processed by manual thresholding at the level required to remove ≥95% of fluorescence in tissue-matched, secondary antibody-only control images[37]. Alongside, the reflectance data were carefully thresholded at the level required to remove all signal from the images collected from the negative control group mice (e.g., fed the AIN-67 diet alone; strategy shown, Supplementary Fig. 1). Thresholded reflectance and fluorescence intensity information per cell or TLV object, alongside size and shape features were then measured for both the immunofluorescence and reflectance channels using the MeasureObjectSizeShape and MeasureObjectIntensity modules in CellProfiler. The number of TLV objects per cell was measured using a RelateObjects module. Data were pre-processed by discarding objects outside of the 5th and 95th percentiles by size prior to analysis as is recommended best-practice[59].

## Cell segmentation: block processing tilescans

To avoid memory limitations, when necessary, tilescanned images were cut into smaller pieces, run through one of the CellProfiler pipelines before reassembling the object segmentation masks and assigning global cell position coordinates to the extracted object features. This was achieved using two MATLAB functions developed in our previous work[37]. The first TilescanToCellProfiler, reads stitched images in most microscopy formats and cuts them into manageable tiles with edge-overlap for processing. The second, CellProfilerToTilescan reassembles the object masks, removes double hits on overlap edges and assigns global position coordinates values to the extracted features. Example images and MATLAB code demonstrating these functions are available for download at the BioStudies database under accession number S-BSST875.

## Cell segmentation: accuracy assessment by Jaccard index

The Jaccard index (intersection over union) approach was used to check the accuracy of the cell segmentation results used in this work. This was achieved by comparing pixel positions inside the automatically-segmented cell objects against those obtained by manual annotations performed by an experience cell biologist. The Jaccard index was calculated as:

$$J(A, M) = \frac{|A \cap M|}{|A \cup M|} = \frac{|A \cap M|}{|A| + |M| - |A \cap M|} \quad (1)$$

Where $J$ is the Jaccard distance for two sets of pixel positions for the automated ($A$) and manual ($M$) segmentations, respectively. Scores of zero represent no overlap (false negatives) whereas scores of 1 represent exact pixel-for-pixel intersection. In this way, it is acknowledged that this approach is a relatively harsh success measure, and that scores of ~0.7 indicate a good segmentation result[57]. This is partially due to the inevitable inaccuracies that are present—even in the manually-annotated data (e.g., due to outline smoothing, ambiguity in determining the precise position of each cell's boundary from the fluorescent staining information, and the available image resolution, etc.).

## Cell segmentation: cell mapping visualisations

The cell map visualisations displaying the delivered cellular doses of fgTiO$_2$ or TLV counts per-cell were produced using cell feature information extracted by the CellProfiler MeasureObjectIntensities and RelateObjects modules alongside the object segmentation masks outputted by each pipeline. Using MATLAB scripts, object feature data were binned into categories (e.g., low, intermediate and high fgTiO$_2$-load based on the integrated intensity of the thresholded reflectance signal per-cell). Using the segmentation masks, individual cell-objects were then colour-coded according to these categories to provide a tissue map visualisation. Example data and MATLAB code demonstrating the creation of these visualisations are available for download from the BioStudies database under accession number S-BSST875.

## Statistical analysis of image-based data

Integrated intensity per object (i.e., per cell or NLV) distributions were compared using two-sided Wilcoxon rank-sum analysis. This non-parametric approach was chosen for compatibility with unpaired samples with unequal group sizes[37]. The approach tests the null hypothesis that the distributions under comparison might reasonably be drawn from one continuous distribution with equal medians ($P > 0.05$) versus the alternative hypothesis that the distributions are distinct ($P \leq 0.05$). Pairwise comparisons of cell counts were first checked for distribution normality and variance homogeneity using Shapiro-Wilk and Bartlett tests, respectively. Where these tests were passed ($P > 0.05$), groups were compared using a two-sided, unpaired samples T-test assuming equal variance.

## Olink® proximity extension assay

Frozen mouse ileal samples were weighed and stored in Kimble pellet pestle tubes (DWK Life Sciences, K-749520-0000) at −70 °C until processing. The Peyer's patch-enriched samples were first trimmed tightly to either side of Peyer's patch using a cold safety razorblade such that each sample primarily consisted of only the patch itself. Tissues were lysed using the BioPlex Cell Lysis Kit (BioRad, 171-304011). 500 μL of Cell Wash Buffer per sample was used for rinsing the tissues. Cell Lysis Buffer was prepared by adding Factors 1 and 2 as per the manufacturer's instructions, as well as Halt Protease Inhibitor Cocktail (ThermoFisher, 78437; 10 μL of cocktail per 1 mL of buffer). Following a rinse, tissues were homogenised by adding 500 μL of ice-cold Cell Lysis Buffer and hand grinding on ice using disposable pestles (DWK Life Sciences, 749520-0000; 20 strokes). Homogenised samples were frozen at −70 °C for at least 4 h, thawed on ice, sonicated in an Ultrawave U300 ultrasonic

bath (44 kHz, 35 W) on ice for 40 seconds and centrifuged for 5 min at 6000 × $g$. Supernatant was collected into 2 mL cryogenic vials (Corning, 430659) and frozen at −70 °C until quantification. Quantification was performed using the DC Protein Quantification Kit I (BioRad, 5000111) as per the manufacturer's instructions (Microplate Assay Protocol). Plates (Greiner Bio-One, 650185) were read using FLUOstar Omega microplate reader at 750 nm and protein concentrations were calculated by the Omega's MARS Data Analysis software using linear regression fit. All samples were normalised to 0.77 mg/mL (+/− 0.12) by dilution with Cell Lysis Buffer and re-quantified to confirm protein concentration. A microplate was prepared for the proximity extension assay by transferring 40 μL aliquots of all samples, negative controls (Cell Lysis Buffer), duplicates of 6 samples, and dilutions (1:4, 1:8 and 1:16 v/v) of two samples into a 96-well skirted plate (ThermoFisher, AB0800) in a randomised manner (controls presented, Supplementary Fig. 11). The plate was sealed (ThermoFisher, 4306311) and stored at −70 °C until shipment on dry ice to Olink (Uppsala, Sweden) for analysis using the Target 96 Mouse Exploratory panel (www.olink.com/mouse-exploratory). All protein names and abbreviations are listed in Supplementary Table 1. Statistical analysis was performed using the Olink Analyze 3.6.0 R-package (https://CRAN.R-project.org/package=OlinkAnalyze). Treatment groups were compared using a two-sample Mann-Whitney U test with correction for multiple testing using the Benjamini-Hochberg method. In the main text, results for proteins that could be involved in inflammatory signaling are shown (Fig. 5) whereas in the Supplementary Information results of the entire protein set are shown alongside repeat-sample, serial dilution and blank sample proximity extension assay controls (Supplementary Figs. 11-13). The proximity extension assay data are available for download at the BioStudies database under accession number S-BSST875.

## Enzyme-linked immunosorbent assay for *Salmonella*-specific IgA

During Mouse Study 2, faecal pellets were collected pre-study and at the 16-week +3 day and 16-week +28 day time-points. Samples were weighed and diluted 1:10 (w/v) in 10% PBS containing 1 mM EDTA. The resulting suspension was vortexed for 15 min to facilitate liquefaction then centrifuged at 400 x $g$ at 4 °C for 5 min. The supernatant was aspirated and centrifuged for a second time at 12,000 x $g$ for 10 min at 4 °C. *Salmonella*-specific IgA quantification was then carried by sandwich ELISA method. Plates were coated with 50 μL lipopolysaccharide (LPS) (Sigma, L2262) diluted in Reggiardo's buffer (0.05 M glycine, 0.1 M NaCl, 1 mM EDTA, 0.05 M NaF and 0.1% sodium deoxycholate). Plates were then sealed and incubated overnight at 37 °C. Next day, the plates were washed 3 times with PBS-tween and blotted dry. Blocking buffer from the whole IgA kit (Affymetrix, 88-50450) was added to each well for 1 h at 37 °C. The faecal supernatant was then diluted in kit buffer A (Affymetrix, 88-50450) and 50 μL of diluted sample was added to each well. After 3 h of incubation, the plates were washed three times with PBS-tween before addition of 50 μL horseradish peroxidase-conjugated IgA. Three more washes were carried out before the addition of 50 μL tetramethylbenzidine substrate for 10 min. This was followed by 50 μL of stop solution prior to absorption reading at 492 nm on a plate reader. All outputs were calibrated using a standard curve and corrected for dilution.

## Faecal ICP-MS analysis

Faecal samples were collected and stored at −80 °C. Samples were weighed and a digestion solution of 1:1 (vol/vol) nitric acid (Fisher, 7697-37-2) and hydrogen peroxide (Sigma, 7722-84-1) was added to the samples (20−70 mg faecal weight) at a 1:10 (wt/vol) ratio. Samples were digested at room temperature for 72 h before loosely-capped sample containers were transferred to a water bath (40 °C) for 5−6 h to release the peroxide. Samples were then transferred to a PTFE vial and sulphuric acid was added (1:1 vol acid/sample weight) prior to final digestion using a Milestone UltraWave microwave. The resultant liquid was decanted

and diluted 1:10 (vol/vol) using ultra-high purity water. Quantification of elemental Ti was performed using an 8900 triple quadruple inductively coupled plasma mass spectrometer (Agilent, USA). Scans were performed in MS/MS mode with the reaction cell adjusted for analysis in $H_2/O_2$ gas. The instrument was set up to identify the mass pairs 48 and 64 for isotopic Ti-48 and $TiO_2$-64. Calibration standards were prepared in 2% nitric acid and spiked with elemental Ti (Sigma, 12237) to final concentrations between 0 and 1 ppm. All samples were spiked with an internal standard solution containing gallium (Sigma, 16639), germanium (Sigma, 05419), yttrium (Sigma, 01357) & europium (Sigma, 05779) at final concentrations of 2 ppb. Counts per-second (cps) values for yttrium were collected to reference signal stability. A 2% solution of nitric acid was used as sample carrier and rinse.

## Reporting summary

Further information on research design is available in the Nature Portfolio Reporting Summary linked to this article.

## Data availability

The image-based cell profiling data generated in this study have been deposited in the BioStudies database under accession code S-BSST875. All data underlying this study are available from the corresponding author upon request.

## Code availability

MATLAB code (using Deep Learning, Image Processing and Computer Vision toolboxes) and CellProfiler pipelines enabling reproduction of the presented image analysis workflows are downloadable from the BioStudies database under accession code S-BSST875.

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

## Acknowledgements

The authors acknowledge the UK Medical Research Council (grant number MR/R005699/1, awarded to J.J.P.), the UK Engineering and Physical Sciences Research Council (grant EP/N013506/1, awarded to H.D.S.), the UK Biotechnology and Biological Sciences Research Council (grant number BB/P026818/1, awarded to P.R.) and the Swedish Research Council (Grant No. 2021-02566, awarded to J.D.S. and Å.V.K.). Mouse Study 1 was funded by the Riddet Institute through its Centre of Research Excellence funding (awarded to N.C.R. by the New Zealand government). Additional funding was provided by the Ministry for Science and Innovation contract C11 × 1009 through Nutrigenomics New Zealand, a collaboration between AgResearch, Plant and Food Research, and the University of Auckland (awarded to D.O.). J.W.W. is grateful to Girton College and the University of Cambridge Herchel-Smith Fund for supporting him with post-doctoral Fellowships. S.R. was supported by doctoral scholarships from Massey University. J.W.W. thanks D. Edwards for her assistance with the microscopy studies, G. Dew for his comments on the manuscript and S.T.E and M.C.E for their support. The authors would also like to thank Prof. N. Mabbot and Prof. S. Milling for their helpful comments. The authors are grateful to Dr. A. Grant for his input into the study design for Mouse Study 2.

## Author contributions

J.W.W. and J.J.P. conceived the concept and designed the analyses. N.C.R. obtained funding for S.R.'s Ph.D. studies including Mouse Study 1. S.R., N.C.R. and D.O. carried out Mouse Study 1. R.J., A.B.D.S., M.M., C.A.P.B., J.W., P.M. and J.J.P. carried out Mouse Study 2. Å.V.K. and J.D.S. collected the human surgical specimens. J.W.W., J.R., M.M., A.F. and R.E.H. conducted the light microscopical studies. A.D., M.M., J.W.W. and J.J.P. performed the protein expression analyses. A.P.B., S.M. and J.W.W. carried out the electron microanalytical work. J.W.W., P.R., H.D.S. and J.J.P. performed the data analysis. J.W.W., P.R., H.D.S. and J.J.P. wrote the manuscript in close collaboration with all authors.

## Competing interests

J.W. states that after completing the presented work as University of Cambridge Herchel-Smith Fellow, he subsequently undertook employment with the pharmaceutical company, GlaxoSmithKline. A.D., J.R., M.M., S.R., H.D.S., R.E.H., A.F., A.B.D.S., C.A.P.B., S.M., Å.V.K., J.D.S., N.C.R., D.O., R.J., P.M., A.P.B., P.R. and J.J.P. declare no competing interests.
