## [Transparent Peer Review file · Nature Communications]

Immunocompetent Cell Targeting by Food-Additive Titanium Dioxide

Corresponding Author: Dr Jonathan Powell

Version 0:

Reviewer comments:

Reviewer #1

(Remarks to the Author)

In this manuscript, the authors investigate the intestinal uptake of titanium dioxide particles. They first show that these particles are internalized mainly in Peyer's patches, but not in the villi, in both humans and mice. They then analyzed the cells responsible for this uptake. They show that the particles accumulate mainly in the SED and at the base of the follicles in autofluorescent macrophages called Lysomacs and pigment cells, respectively. This is an extremely well-documented observational microscopy study, but unfortunately the consequences of this uptake were not addressed. Overall, this study is mainly descriptive, with nice imaging but no new functional data. Furthermore, it is already known that microparticles are mainly internalized by Peyer's patch phagocytes, especially macrophages. In the absence of functional data, I would recommend submitting this manuscript to a more specialized journal than Nature Communication, where we expect more in-depth studies.

Major concerns:

As mentioned above, there is no functional approach carried out in this study and there are many questions that remain unanswered:

1/ what are the kinetics of uptake/clearance of titanium dioxide particles? How long do particles remain in the SED after a chase? Do some particle-loaded cells migrate to other regions of PP? What is the ratio of SED to basal accumulation over time?

2/ Although the authors claim that particles are only taken up by Lysomacs, this is based on the autofluorescence signal only and no specific markers were used. Furthermore, in their Figure 4p to 4s, different levels of autofluorescence can be seen among particle-loaded phagocytes, suggesting that some weakly autofluorescent phagocytes may also take up particles. These data need to be confirmed. From the literature, it appears that after M cell transcytosis, luminal particles are internalized by Lysomacs, but also by another phagocyte subset, termed LysoDC, which is less autofluorescent. The authors should use the specific markers that distinguish the different Peyer's patch phagocyte subsets to determine whether only Lysomacs are able to internalize fgTiO₂ or whether LysoDCs or even conventional DCs can also do so. This is important because macrophages do not mount an adaptive immune response, whereas DCs do.

3/ In this context, is there a change in the immune response to food antigens after ingestion of titanium dioxide particles? This could be done by adoptive transfer of OVA-specific T cells and analysis of proliferation and polarization. Also, is there a change in the distribution, location and activation of phagocytes, B and T cells after ingestion?

Minor concerns:

Introduction: the introduction focuses only on fgTiO₂ particles with no reference to Peyer's patches other than their ability to store these particles, and the terms SED, M cells, Lysomacs are used in the Results section without any prior description. A short paragraph on the functions, organisation and cellular composition of Peyer's patches in the Introduction would facilitate the reader's understanding.

I. 115 to I. 117: The authors point out that the specificity of particle uptake by Peyer's patches versus villi is unclear. This statement has to be qualified as the work of Neutra's team in the late 1990s (Frey et al, JEM, 1996; Mantis et al, AJP, 2000) showed the selective adhesion of nanoparticles over 30nm in diameter to Peyer's patches, mainly due to the protection of the villus epithelium but not the FAE by the glycocalyx coat. In addition, many groups have shown preferential uptake of microspheres in Peyer's patches, particularly by M cells, LysoDCs and LysoMacs. This should be recognized.

Figure 2k: In this figure the particle appears to be outside the GP2-labeled cell, i.e. in a follicle-associated enterocyte.

Staining of the basolateral membranes (e.g. EpCAM staining) in addition to GP2 would help here to clearly delineate the

boundary between the M cells and the enterocytes.

Reviewer #2

(Remarks to the Author)

The submitted paper tackles the important issue of the absorption of TiO₂ in the intestine, since this additive has been banned from food only by EFSA, but is still present in other part of the world, as well as in drugs. The authors employ very elegant techniques to localize the TiO₂ particles in the ileum and caecum of mice with a chronic TiO₂ consumption, but they also evaluated their distribution in human samples obtained from surgery. The obtained results, derived from a combination of immunofluorescence, confocal microscopy, confocal-electron microscopy and energy dispersive X-ray analyses, localized the TiO₂ particles within Peyer's patches in the ileum, in particular within lysomac cells present in the immunocompetent SED region.

Although the paper provides important information regarding the interactions between TiO₂ particles and cells of the immune system, a more detailed analysis of the characteristics of the lysomac cells loaded with TiO₂ would add a lot of impact to the paper

The authors should compare their results with those previously reported in the literature (although obtained with different methods), in particular regarding the role of the epithelium (for example <https://doi.org/10.1186/s12989-020-00357-z>)

Reviewer #3

(Remarks to the Author)

Foreword: My background is purely computer science, image processing. I was asked by the editor to review the image processing of this manuscript.

My understanding of the whole setting is the following. The goal is to detect and quantify the presence of fgTiO₂ additive in Peyer's patches (certain intestinal cells) in human tissue. The tissue was imaged using light and electron microscopies and X-ray spectroscopy to confirm that in the reflectance confocal microscopy the additive displays as bright foci, the brightness of which can be quantified subsequently. This confirmation has been reached qualitatively.

Furthermore, two groups of mice, those fed also with fgTiO₂ and those that were not, were used to confirm this co-localization finding; and also to set up the image processing pipeline and, foremost, to confirm the scientific finding.

The whole analysis is very much dependent on the outcomes of the image processing, making it a critical step in the whole study. Notice that tiny particles, which are hard to find for humans, are sought in the images and their area is computed. The overall area is rather small making every mis-detected pixel relatively influencing the overall computed outcome. I would, therefore, double-check every step of the pipeline, and would subject it to dedicated testing, and report this as well (to allow the readers to gain trust in the conducted measurements). This attitude is driven by my experience that image processing, including deep learning-based one, is in general very data sensitive, and requires fine-tuning to the particular data. Taking any pipeline, even when it has been published previously for the same or similar-looking data, does not necessarily guarantee any quality and accuracy of the calculations.

For example, Fig. 2b-d goes nicely in this direction by trying to question the sensitivity of the foci detection. Nevertheless, I haven't understood from the main text (L130-139), the figure, or its caption what is displayed in 2d, and why I see a bright spot in 2c (if it's supposed to be after the milling) in the white region, and why is there the yellow region. Another step towards "gaining trust" in the detection would be to show a small number of insets with the original image without the translucent markers (before the markers were added) in the figures where the markers are used. Simply to see what the data looked like and to decide for myself if the marker was really where it should be.

In general, the Authors have done a great job of reporting how were images processed. I think I see the whole picture and understand (from the text, from the supplementary as well as from the EMBL-EBI-published code and data) what steps were taken and in which order. I have downloaded the EBI .zip file and familiarized myself with the available images. And I became a believer that the reported image processing could indeed work well on the data. However, the devil is in the details. And so I wrote down remarks that, if they would have been clarified in the manuscript right away, I would have been left without any doubts about the image processing used in this study.

I think that the reported image processing is supporting the biological finding. That said, even if some details of the processing would have been carried out slightly differently, the overall outcome would continue supporting the biological claims.

Remarks:

During the processing of the murine tissues, I understand and accept the plan that, given the fixed experiment and imaging conditions, the threshold level is determined on the non-fgTiO₂ images and then used in the fgTiO₂-containing images to detect it. However, when switching to human tissues, is it guaranteed that the fgTiO₂ signal appears in the same proportion compared to the rest of the tissue signal so that the same threshold level can reliably detect fgTiO₂. I am wondering if, for example, cell density is the same between the murine and human tissues (assuming the cell density may influence the propagation and thus strength of the wavelength).

Was there one (ideal) threshold that worked very well across all non-fgTiO2 murine tissues? From the opposite side, I had a look at 'C4_section1.tif - C4_section1_3 - C=3.tif' file from the Figure 1 folder from the EMBL-EBI public data, and I see that the loci show themselves as clear (and high enough) intensity peaks rendering the (global) thresholding as indeed a viable technique. However, a range of threshold values was giving reasonable results, and with decreasing threshold was increasing the number of detected (and still rather reasonable looking) loci. The threshold interval is established from non-fgTiO2 images. However, an interval of permissible thresholds (everything beyond a minimal threshold that "wipes out" everything) could be easily established for the non-fgTiO2 images, opening a room to choose the final threshold, but this room could influence the number (and size) of the detected fgTiO2 loci.

Considering Figs. 1 and 2, Supplementary Fig. 2, the lysomac cells were acquired as 3D stacks while the reflectance images seemed to be 2D, how was the matching slice from the 3D stack selected in order to measure, e.g., the foci vs. full cell area (the cell dosimetry)?

Since the confocal images were tile-scanned, with overlap, and then presumably registered and stitched, I understood that the lateral coverage of the reflectance images was obtained similarly, thus also by tile-scanning. How were the reflectance images registered before the stitching? Is it ruled out that the stitching process could introduce false positive foci?

How were the deep learning models trained? Especially, how were the training data prepared? If, for example, a network was classifying reflectance foci into three density classes Lo/Int/Hi, it must have been trained to do so. Was the classification done by one expert, or a consensus among experts (at least four to warrant any class is chosen at least two times)?

For the single-cell dosimetry measurements, the above set of questions applies also to the segmentation network that was used. Additionally, what level of values were there usually found in the segmentation probability maps? The Otsu thresholding cares about modes in the histogram, even if the modes are low (uncertain).

For the employed watershed, what seeds were used and how were they determined?

Note about code availability: The published code (from the EMBL-EBI site) is mostly a code to produce the figures themselves, not so much a code to conduct the analyses.

PS: I think there is a systematic typo visualisation -> visualization.

Vladimir Ulman

Version 1:

Reviewer comments:

Reviewer #1

(Remarks to the Author)

I appreciate the authors' efforts to improve the quality and impact of their data. Throughout their revision, they carried out several experiments with specific objectives:

1/Confirming the identity of phagocytes responsible for capturing fgTiO2 particles, which was crucial. Their effort revealed that fgTiO2 were not only captured by macrophages (LysoMacs and pigment cells) but also by LysoDCs, thus addressing this point adequately.

2/Establishing the persistence of fgTiO2 particles in Peyer's patches, especially in the subepithelial dome (SED) region, and their dispersion in the tissue post-infection.

3/Identifying any immunostimulatory activity of fgTiO2 particles.

However, despite these efforts, the newly acquired data are relatively minor, with the exception of identifying LysoDCs as an additional phagocyte subpopulation internalising TiO2 particles and observing the persistence or dissemination of TiO2 particles in the SED of PP at homeostasis or during infection, respectively. Without compelling functional assays, I maintain my belief that the authors should consider submitting their interesting yet descriptive work to a more specialized journal.

Major points:

Authors' first objective was addressed successfully, although for formal identification of cells (e.g., determining if all TiO2+ cells with high autofluorescence express high CD4 but low MHCII levels?), it would have been preferable to display all markers and signal detections, i.e., TiO2 particles, auto-fluorescence (AF), CD4 and MHCII, on the same cells instead of separately on different cells. Nevertheless, their complementary experiments seem to indicate that LysoDCs (AFint and CD4- and MHCIIhi) also capture TiO2 particles.

Given the pivotal role that LysoDCs play in T cell priming, it is crucial to explore whether the uptake of TiO2 particles influences the activation (e.g., surface expression of MHCII and co-stimulatory molecules after incubating sorted LysoDCs with TiO2 particles, compared with known LysoDC stimuli) and priming capacity of LysoDCs (e.g., co-culture experiments with T cells). Despite the availability of well-established methods for such analysis, which have been previously published by different research groups and do not necessitate complex procedures, this aspect has not been investigated. Consequently, it remains challenging to conclusively state, as claimed in the abstract, that TiO2 particles lack adjuvant

effect. Incorporating these functional assessments would greatly enhance the study.

It is unfortunate that cellular markers were not used to assess the persistence of TiO₂ particles in the SED. Without such markers, it becomes challenging to definitely identify cells containing TiO₂+ vesicles at 28 days. Specifically, while the authors have noted a higher number of TiO₂+ vesicles per cell at 28 days, it remains unclear whether these cells are the same than those observed at 3 days. Given the different lifespans of LysoDCs and LysoMacs, the possibility of secondary capture by long-lived phagocytes should be envisaged. However, in the absence of markers it is impossible to draw any conclusion beyond a change in particle localisation post-infection.

The experimental approach used to assess the adjuvant activity of TiO₂ particles presents an issue as the influence of TiO₂ particles on the expression of inflammatory proteins is consistently examined alongside an infection using a non-virulent auxotrophic strain of Salmonella, known for its effectiveness as a vaccine. Considering the robust adjuvant activity demonstrated by this strain in numerous publications since the 1980s, it is unlikely that a specific effect of a potential adjuvant would be distinguishable in co-administration. Therefore, the relevance of such an experiment is limited, as multiple immune pathways are already activated by the attenuated bacteria via various PAMPs such as LPS, flagellin and bacterial DNA, among others. It would have been more pertinent to evaluate the effect of TiO₂ alone, in comparison to mice without TiO₂, within a model of tolerance versus classical immunisation.

Specific and minor points:

Figure 5f: this panel of the figure is very difficult to look at (multitude of very small panel with almost overlapping text). Perhaps the authors could consider selecting the most relevant molecules for display and relocating the others as supplementary data. Framed text is usually used to emphasize important elements of the figure; however, here, it is used to highlight analytes that are at least in part below the detection threshold. This is misleading. I would suggest to remove this frame.

Figure 6: As mentioned above, incorporating cell markers such as CD11c, CD4, MHC-II and lysozyme would greatly improve the meaning of the data by providing an identity to TiO₂-loaded cells at day 28 compared to day 3.

Figure 6a-b: I would suggest to add autofluorescence here to highlight the difference with panel L.

Figure 6c-d: It seems that there are already many TiO₂-loaded vesicles in the follicle at day 3, possibly even more than in the SED. Moreover, by day 28, these TLVs appear mostly located in the midpart of the follicles rather than at the basal part. To further elucidate these observations, quantifying the TiO₂-loaded vesicles according to specific regions, similar to the approach used in Figure 2n, would provide valuable quantitative data and enhance the analysis of the spatial distribution of these vesicles over time.

Figure 7a: the quality of these images is rather poor making it challenging to visualize individual villi and even the lumen in the last picture. Improving the image quality would enhance clarity and facilitate a more precise interpretation of the data.

Result section:

I. 264: ref 27, Monack's lab work is anterior to the characterization of LysoDC's in the early 2010s.

I. 265-266: Several studies spanning several decades have demonstrated not "modest" but rather robust adjuvant activity of the attenuated Salmonella strain. Although this strain does not induce disease as it is rapidly eliminated, it is known to elicit potent humoral and cellular immune responses.

I. 303: this sentence may be misleading as it could imply the absence of macrophages in the SED at day 28, which is unlikely. It would be beneficial to stain macrophages (e.g., MerTK and Lysozyme) in this context to provide clarity. The most plausible explanation is that the renewal of macrophages following infection likely decreased their autofluorescence, as newly recruited macrophages may have a reduced phagocytic history compared to those they replaced.

I. 310-I. 315: without labeling with LysoDC and LysoMac specific markers and excluding villus lamina propria macrophages, there is insufficient evidence to support this scenario. Therefore, it would be more appropriate to move these sentences to the discussion section. Additionally, recent studies have demonstrated that indeed stimulated LysoDCs migrate, albeit to the IFRs rather than the villus lamina propria.

(Remarks on code availability)

Reviewer #2

(Remarks to the Author)

The submitted paper analyzes the fate of food-grade titanium dioxide in the intestine of mice identifying, in the subepithelial dome region of Peyer's patches, LysoMAC and LysoDC cells as the final destination of this additive. This observation obtained in mice was confirmed by the analysis of human surgical samples, where titanium dioxide was localized in the same cell types. Due to the active role of DC cells in the immune response, the authors evaluated if the presence of titanium dioxide was able to alter the response to attenuated Δ aroA-Salmonella, which colocalized in the same cells. The

assessment of the level of several proteins involved in immunity and inflammation did not show any difference in the animal previously treated with titanium dioxide compared to those on normal chow.

The data presented in the paper are providing convincing evidence regarding the localization of the titanium dioxide particles in the Peyer's patches. However in the discussion the authors should underline that the lack of effect of titanium dioxide on the immune response is strictly limited at model employed in this study. In addition, in their experimental design the administration of titanium was suspended before that of the Δ aroA-Salmonella. Due to the fact that titanium dioxide is a food additive, it would be quite important to assess if its interaction with food components can alter their presentation to the intestinal immune system.

Moreover, although the authors did not show any accumulation of titanium dioxide in the epithelium they cannot exclude any functional damage to enterocytes, and this should also be taken into account. Last but not least, the lack of additive effect cannot be excluded in condition affecting the intestine, such as IBD or celiac disease.

(Remarks on code availability)

Reviewer #3

(Remarks to the Author)

I would like to thank the Authors for carefully and patiently answering all my questions, and I very much appreciate that they conducted additional experiments and analyses also towards strengthening the description of the image processing and analysis parts, and that they cared to describe everything in the manuscript. Also thanks for the information and all three figures in the rebuttal letter.

The new insights into the pipelines and pokes into the intermediate results make the whole study a lot stronger for me. I have to say that the newly reported values are very good, and convincing.

For example, the new Supplementary Figures 2 and 5 are both nicely illustrating the data, the image processing problem, and the typical outcomes. Besides, Supp. Fig. 2 shows additionally very good segmentation accuracy!

Regarding the settings of the threshold: It is true that previously I haven't seen the connection that the given biological-physical-optical setup could warrant that the distributions of image intensities across both human and mouse images can be very similar, and thus the threshold value established in the first group of images can be carried over to the second group of images. I thank the Authors for reminding me, and especially thank them for providing the new Supplementary Figure 4 that demonstrates the situation perfectly. (And there's no need to quantify the observation any further.)

These two topics above were previously my two greatest concerns about the methodology, which are now cleared away.

The updated descriptions of the image processing, in the Methods section, are now clearer to me too because they contain now more of a technical context and reasoning to justify the steps taken. I find this new part of the manuscript to be far more easily reproducible than what it was before. And I again thank the Authors for their effort on this.

The publicly available data on the BioImage Archive are findable, accessible, interoperable, and seem reusable (upon random digs into the source codes and images), very nicely organized now (thank you again).

A very minor remark would still be left, though. In the new Supp. Fig. 13, the white rectangle in panel c) is not echoed elsewhere in that figure (and is also not mentioned in the caption); could it be because of the white frame on a white background? And there's also maybe(?) a mismatch in the descriptions as the b) segmentation of nuclei looks more like a segmentation of a, top) the actin channel rather than a, bottom) the nuclei channel; the descriptions in panel a) are probably swapped. And could the segments boundaries be drawn in e.g. green (not black) in panel d) to facilitate qualitative demonstration of the watershed performance? (It's okay and normal if the performance is not superb everywhere.) Similarly, the segments' contours in [Supp. Fig. 14 c), two right most] and [Supp. Fig. 15 f, g] are suggested black when the actual contours are white; could they be e.g. green like they already are around?

Congratulations on such a nice work!
Vladimir

(Remarks on code availability)

I haven't checked all the code that's now available, I checked only randomly some files. But I can conclude that the code+data is well organized, and the source code is understandable. It is a mixture of Matlab, Python, and CellProfiler scripts, but that's okay. I haven't tried to execute anything.

Version 2:

Reviewer comments:

Reviewer #1

(Remarks to the Author)

As this is the second round of review, I will only respond to the points I previously raised that have not been addressed and the reasons provided by the authors for this, or for which I still have concerns. Other points have been adequately addressed by the authors and I thank them for that.

Authors' response: Notwithstanding, we have considered the reviewer's suggestion very carefully concerning a potential T cell adjuvant effect and taken advice from colleagues with expertise in the area. Collectively, it would not be trivial to extract and purify sufficient LysoDC cells from Peyer's patches, without activating them in the process, and to load them with particles / antigen in a way that mimics the in vivo setting and look at markers of adjuvanticity. 'If properly characterised and validated, it is probably the work of a PhD in an expert lab' was some feedback that we received.

My comments on this response: I understand that if none of the authors have expertise in in vitro culture of phagocytes, such experiments may seem challenging. However, other straightforward functional tests could have been performed instead, such as DTH (delayed-type hypersensitivity) response tests to determine at least whether TiO₂ can induce a loss of oral tolerance. This is a critical point to evaluate its innocuity. These types of tests do not require cell extraction but just involve measuring footpad or ear swelling after mice have been fed several times with an antigen (e.g., OVA), immunized with the same antigen and then challenged subcutaneously in the footpad or ear (e.g., PMID: 1653388).

Authors' response: We did, however, address the reviewer's suggestion of looking at particle effects on MHCII and co-stimulatory molecules (CD80/86), specifically in the fgTiO₂ + cell population of the Peyer's patch SED (i.e., LysoDCs/LysoMacs). The data are shown below in Fig. A and reveal no measurable changes in response as a result of in vivo particle uptake. We have not, at this stage, included these results in the manuscript because, as noted above, our focus is very much on inflammatory augmentation. We are of course happy to do so if the reviewer / editor feels including these results would be helpful?

My comments on this response: Unfortunately, changes in MHCII, CD40, CD80 and CD86 surface expression need to be measured by flow cytometry to achieve sufficient sensitivity. Therefore, I suggest excluding this figure, as it is misleading and does not effectively convey the intended information.

Authors' response: In the revised Figure 7A we've improved the resolution and added labels to better indicate the subepithelial dome (SED), lumen and villous mucous (VM) regions in each tilescan. We've also included enlarged versions of these three analyses alongside the fluorescently-counterstained images to provide improved histological context in the new Supplementary Figure 15.

My comments on this response: It is necessary to include the actin staining shown in Supplementary Figure 15 in the main figure, as it provides valuable insight into the tissue morphology. The authors should label all SEDs, not just one (image 2 and 3 contain two SEDs each), and use actin staining (brush border) to accurately delineate the follicle-associated epithelium (FAE). Notably, the FAE border in image 2 is incorrectly marked. Additionally, the term "basal aspect" in the main text is misleading. The authors should adhere to Peyer's patch (PP) nomenclature. Specifically, the PP regions located in between follicles (and SEDs) are termed interfollicular regions (IFRs). These are the regions where part of TiO₂-loaded cells accumulates upon Salmonella infection, distinct from pigment cells observed at the base of the IFRs and follicles. IFRs are where naïve T cells are located and primed, and where LysoDCs, but not LysoMacs, migrate upon stimulation (PMID: 37580335 and PMID: 32268097). This should be clearly mentioned in the manuscript.

Minor points:

L97-99: Given the topic of the study, the authors should clearly mention here that LysoDCs and LysoMacs are specialized for the uptake of dead cells, particulate antigens and pathogenic bacteria, in addition to their role in anti-microbial defense. This is particularly relevant since TiO₂ uptake is the main focus of the manuscript.

L269-272: The phrase "resulting in a modest immuno-inflammatory response (unlike bacterial fragments, to which the intestine is uniquely tolerant)" should be removed, as the immune response to this strain is strong enough to induce long-term protection against the virulent strain, and thus cannot be considered "modest".

L329-330: Rather than depleting the SED region of previously absorbed plastic microparticles, oral danger signals, such as Salmonella, induce the migration of antigen-loaded LysoDCs from the SED to the IFRs (PMID: 12496201, PMID: 37580335 and PMID: 32268097). This needs to be specified.

L333-334 and 371-372: There are several hypotheses that could explain the presence of TiO₂ in the villi during Salmonella infection. Regardless of the type of stimulation, including Salmonella infection, LysoDCs have never been observed in the villus lamina propria. Therefore, if the authors wish to support the hypothesis of LysoDC migration into villi, they must at least demonstrate that these villus TiO₂-loaded cells exhibit the phenotype of LysoDCs. This includes showing lysozyme expression and the absence of CD4, as well as the lack of the villus macrophages markers CD64 and F4/80. Autofluorescence alone is characteristic of all macrophages and cannot be used to define LysoDCs, which are actually less autofluorescent than LysoMacs.

If such evidence is not provided, these lines should be removed, as it is not supported by the data.

(Remarks on code availability)

Reviewer #2

(Remarks to the Author)
no further comments

(Remarks on code availability)

Version 3:

Reviewer comments:

Reviewer #1

(Remarks to the Author)

During three rounds of revision, I suggested several experiments to extend the functional aspects beyond the authors' initial observations. These have been only partially addressed at best. However, I acknowledge that some of these experiments—though not all—were technically challenging. Overall, I appreciate that the manuscript provides interesting insights and establishes a new mouse model to assess the health effects of ingested nanoparticles.

(Remarks on code availability)

Response to the reviewers' comments –

The authors are grateful to the reviewers for the helpful feedback. Harnessing a new 20-week animal study, **our revised submission contains significant new functional data**. We have also added substantial new results to the Supplementary Information that better explain and demonstrate the accuracy of the image analysis methods. Alongside, the 'BioStudies' data submission accompanying the manuscript has also been completely overhauled to provide all data and downloadable, running examples of the analyses.

A point-by-point response to all comments is provided below, with the positions of all changes corresponding to the marked-up manuscript **highlighted in yellow**.

Reviewer One	
Reviewer's comment	Response to the reviewer
Major comments	
In this manuscript, the authors investigate the intestinal uptake of titanium dioxide particles. They first show that these particles are internalized mainly in Peyer's patches, but not in the villi, in both humans and mice. They then analyzed the cells responsible for this uptake. They show that the particles accumulate mainly in the SED and at the base of the follicles in autofluorescent macrophages called Lysomacs and pigment cells, respectively. This is an extremely well-documented observational microscopy study, but unfortunately the consequences of this uptake were not addressed. Overall, this study is mainly descriptive, with nice imaging but no new functional data. Furthermore, it is already known that microparticles are mainly internalized by Peyer's patch phagocytes, especially macrophages. In the absence of functional data, I would recommend submitting this manuscript to a more specialized journal than Nature Communication, where we expect more in-depth studies.	We appreciate the review and positive feedback on the quality of the manuscript. We have also taken on board the reviewer's suggestion for further substantial ('functional') data on ingested, food-grade titanium dioxide (fgTiO₂) and, therefore, additionally now report on findings from a new 20-week animal study. As such, we now characterise the phenotype of the fgTiO₂-recipient cells more closely, investigate the particles' potential to adjuvant cell responses to a microbial pathogen across multiple pathways and shed significant light on the lifecycle and migratory pathways of the TiO₂-recipient cells within and beyond the Peyer's patch. We also clarify that this is not a manuscript reporting 'model particle – Peyer's patch' interactions. Rather, it is a multidisciplinary study, specifically designed to address a gaping hole in the science of orally-ingested fgTiO₂ in humans (i.e., where does it go, how does one recreate the human exposure scenario in a mouse model and does fgTiO₂ do anything at human-relevant doses?). These specific questions have global importance as regulatory tensions threaten to destabilise a ~17 billion-dollar industry and jeopardise the future availability of European therapeutics (notably for paediatric and orphan medicines) [1]. Importantly, after decades of uncertainty, the provided work allows any question marks hanging over the safety of persistent oral particles to be addressed (ranging from genotoxicity through to differential responses of distinct genotypes). We believe that such important findings deserve communicating in an equally important scientific platform, namely Nature Communications.

(1) what are the kinetics of uptake/clearance of titanium dioxide particles? How long do particles remain in the SED after a chase? Do some particle-loaded cells migrate to other regions of PP? What is the ratio of SED to basal accumulation over time?	We agree: the narrative needs to move beyond the M-cell. To achieve this, we harness the new 20-week animal feeding study noted above with two harvest time-points demonstrating the natural history of fgTiO₂ in its only known site of accumulation (Peyer's patch / small intestine). We show:  • fgTiO₂ first moves from M-cells to both LysoMac and LysoDC cells (Figure 4 t/u) (please see response to Comment 2 below). • Under standard clean conditions of the modern animal houses, the fgTiO₂-recipient cells are 'long-lived' in the subepithelial dome of the Peyer's patch, remaining for at least 4 weeks after cessation of feeding fgTiO₂ (Figure 6a/b). • Over this time, quantitative image analysis of whole Peyer's patch transverse sections reveals a system of vesicular inheritance whereby certain cells accrue increasing numbers of fgTiO₂-loaded vesicles. This explains how 'pigment cells' at the base of Peyer's patches must form (Figure 6 c-j). • Upon subsequent exposure to a human-relevant oral danger signal (attenuated Salmonella) this movement is greatly amplified. In-line with prior seminal work investigating model particles [2], the subepithelial dome region becomes entirely depleted of fgTiO₂ (and autofluorescent phagocytes) (Figure 6k-L). One cell population (likely LysoMacs) moves to the base of the Peyer's patches whereas the other (likely LysoDCs) migrates out of the Peyer's patch to the lamina propria of the regular villous mucosa taking its cargo of fgTiO₂ with it (Figure 7a-d). • Importantly, this explains and recreates the human fgTiO₂ tissue distribution (new Figure 1c/d) and will allow real risk assessments going forward (discussed, Page 14 line 477).
(2) Although the authors claim that particles are only taken up by Lysomacs, this is based on the autofluorescence signal only and no specific markers were used. Furthermore, in their Figure 4p to 4s, different levels of autofluorescence can be seen among particle-loaded phagocytes, suggesting that some weakly autofluorescent phagocytes may also take up particles. These data need to be confirmed. From the literature, it appears that after M cell transcytosis, luminal particles are internalized by Lysomacs, but also by another phagocyte subset, termed LysoDC, which is less autofluorescent. The authors should use the specific markers that distinguish the different Peyer's patch phagocyte subsets to determine whether only Lysomacs are able to internalize fgTiO₂ or whether LysoDCs or even conventional DCs can also do so. This is important because macrophages do not mount an adaptive immune response, whereas DCs do.	This is an excellent point. Please see the revised Figure 4t/u where use of MHCII and CD4 staining indicates that the autofluorescent, fgTiO₂-recipient cells are indeed most likely a mixture of LysoMac (autofluorescence^{hi}, MHCII^{lo}, CD4^{hi}) and LysoDC (autofluorescence^{int}, MHCII^{hi}, CD4^{lo}) cells as the reviewer suggests. Later in the manuscript we also show how – upon Salmonella challenge – a population of autofluorescent cells typically containing low numbers of reflectant foci migrates to the lamina propria (i.e., likely LysoDCs) whereas the more heavily-loaded, highly autofluorescent cells migrate basally to pigment cell zone (i.e., likely LysoMacs). These results are presented in the new Figure 7 and discussed on Page 15 line 499. The authors thank the reviewer for this suggestion which has provided new insights into the routings of persistent particles within and beyond the Peyer's patch.

(3) In this context, is there a change in the immune response to food antigens after ingestion of titanium dioxide particles? This could be done by adoptive transfer of OVA-specific T cells and analysis of proliferation and polarization. Also, is there a change in the distribution, location and activation of phagocytes, B and T cells after ingestion?	We agree that, in terms of both immunoglobulins and T-cells, it would be interesting to see how fgTiO₂ exposure influences immune responses to food antigen. We also agree that our model can now be used to perform experiments such as this. After decades of uncertainty, it provides the human-relevant exposures necessary to determine whether fgTiO₂ accumulation is indeed benign, or in fact has negative effects on mammalian health (now stated, Page 16 Line 531). While we have not directly addressed the issue of whether fgTiO₂ affects interactions between phagocytes and antigen-specific T-cells, we have cast the net wider covering multiple outcomes and our new data show that same-cell uptake, in vivo, of fgTiO₂ does not affect the functions of SED phagocytes also carrying attenuated Salmonella (new Figure 5). These data were generated for 92 different proteins in multiple cellular pathways, using sensitive and specific proximity ligation analysis. We would therefore not anticipate fgTiO₂-induced changes in the immune response to food antigens.
Minor comments	
(4) Introduction: the introduction focuses only on fgTiO₂ particles with no reference to Peyer's patches other than their ability to store these particles, and the terms SED, M cells, LysoMacs are used in the Results section without any prior description. A short paragraph on the functions, organisation and cellular composition of Peyer's patches in the Introduction would facilitate the reader's understanding.	This is an excellent suggestion. We've added a new paragraph to the introduction (Page 5 Line 125) which reads: “ The exception, as noted above, is the small intestine, in both humans and rodent models, and specifically their large lymphoid follicles termed Peyer’s patches. In absence of afferent lymphatics, the apical surfaces of these follicles possess a unique population of microfold (M) cells that avidly sample the lumen for bulky macromolecules including small micro- and large nano- particles. This material is passed to underlying immune cells in the apical, ‘subepithelial dome’ (SED) region of the follicle which contains recognised populations of monocyte-derived, lysozyme-expressing cells termed ‘LysoMacs’ and ‘LysoDCs’ that are specialised for anti-microbial defence. Overall, the major role of Peyer’s patches is initiating processes for the generation of mucosal IgA and systemic IgG in response to intestinal antigen thereby orchestrating the sequence of events that leads to gut-derived humoral immunity. ” Thanks.
(5) I. 115 to I. 117: The authors point out that the specificity of particle uptake by Peyer's patches versus villi is unclear. This statement has to be qualified as the work of Neutra's team in the late 1990s (Frey et al, JEM, 1996; Mantis et al, AJP, 2000) showed the selective adhesion of nanoparticles over 30nm in diameter to Peyer's patches, mainly due to the protection of the villus epithelium but not the FAE by the glycocalyx coat. In addition, many groups have shown preferential uptake of microspheres in Peyer's patches, particularly by M cells, LysoDCs and LysoMacs. This should be recognized.	We agree it is essential that this pioneering work is recognised. We've added a paragraph to the discussion (Page 15 line 487) which now explains the likely reason for this observation: “Seminal early work from Frey and Mantis showed that the mucus-rich glycocalyx markedly impairs model particle uptake by the regular villous epithelium, whereas the near-absence of these features overlying Peyer’s patches allows direct access of particulates to M-cells. These cells have an extraordinary ability to engulf luminal particles and macromolecules, passing them through to abutting cells at their basolateral membrane. We first confirm, unequivocally, that this uptake mechanism holds true for fgTiO₂ and, next, demonstrate that

	specific SED immuno-competent phagocytes (LysoMacs and LysoDCs) are the direct recipient cells for cargo from the M-cell." Many thanks for pointing this out.
(6) Figure 2k: In this figure the particle appears to be outside the GP2-labeled cell, i.e. in a follicle-associated enterocyte. Staining of the basolateral membranes (e.g. EpCAM staining) in addition to GP2 would help here to clearly delineate the boundary between the M cells and the enterocytes.	This is a good point: the brightness of the GP2 channel was not high enough to clearly see the association of the reflectant foci and GP2 in the original submission. In previous work, we've shown that F-actin staining provides good demarcation of epithelial cell boundaries in frozen sections where excessive mucin binding causes difficulties using membrane markers [3]. Harnessing this, in the revised Figure 2k, we've used the actin information to draw the cell boundary and demonstrate two foci moving within this region that further associate with GP2 (shown, Fig. A below). Thanks. F-actin / GP2 / RL / TiO₂ circle-markers  Fig. A. Reflectant foci transiting through F-actin / GP2-delineated M-cells.
Reviewer Two	
The submitted paper tackles the important issue of the absorption of TiO₂ in the intestine, since this additive has been banned from food only by EFSA, but is still present in other part of the world, as well as in drugs. The authors employ very elegant techniques to localize the TiO₂ particles in the ileum and caecum of mice with a chronic TiO₂ consumption, but they also evaluated their distribution in human samples obtained from surgery. The obtained results, derived from a combination of immunofluorescence, confocal microscopy, confocal-electron microscopy and energy dispersive X-ray analyses, localized the TiO₂ particles within Peyer's patches in the ileum, in particular within lysomac cells present in the immunocompetent SED region.	Thank you for reviewing the work, acknowledging the importance of the issue and for the positive comments on the techniques used. Based on overall feedback, we've added a significant amount of new data to the revised manuscript including studies that more closely characterise the phenotype of the TiO₂-recipient cells, their potential to adjuvant immune responses and unravel their uptake / lifecycle / migratory pathways in mice and humans (described below):
(7) Although the paper provides important information regarding the interactions between TiO₂ particles and cells of the immune system, a more detailed analysis of the characteristics of the lysomac cells loaded with TiO₂ would add a lot of impact to the paper	We agree with the Reviewer. To achieve this, we harness a new 20-week animal feeding study and present a significant body of new results:  • Characterisation of MHCII and CD4 expression demonstrates that both LysoMac and

	LysoDC cells receive fgTiO₂ in the Peyer's patch subepithelial dome (Figure 4 t/u).  • Chasing fgTiO₂ feeding with a human relevant pathogen (low virulence Salmonella) shows Salmonella markedly co-accumulates in fgTiO₂-recipient cells (Figure 5a-c). • Study of 92 cell regulatory and inflammatory proteins showed no measurable adjuvant activity of fgTiO₂ on the Salmonella infection – despite heavy loading into the same cells (and previous in vitro results to the contrary) (Figure 5, e/f). • In absence of Salmonella, the fgTiO₂-recipient cells are 'long-lived' in the subepithelial dome of the Peyer's patch: remaining for at least 4 weeks after cessation of feeding fgTiO₂ (Figure 6a/b). • Over this time, there is a system of vesicular inheritance whereby certain cells accrue increasing numbers of fgTiO₂-loaded vesicles. For the first time, this explains how 'pigment cells' at the base of Peyer's patches form (Figure 6 c-j). • Whereas cellular co-accumulation of Salmonella and fgTiO₂ did not appear to adjuvant immune responses, a marked cell migration effect was observed: • The subepithelial dome region of the Peyer's patches was entirely depleted of fgTiO₂ (and autofluorescent phagocytes) (Figure 6k-L). The LysoMac cell population moved to the base of the Peyer's patches whereas the LysoDC population migrated out of the Peyer's patch to the lamina propria of the regular villous mucosa taking its cargo of fgTiO₂ with it (Figure 7a-d). • Collectively, these new findings characterise the identity, lifecycle and migratory pathways of the fgTiO₂-loaded cells in a way that enables complete recapitulation of the human exposure scenario in a mouse model (revised discussion, Page 15 line 487).
(8) The authors should compare their results with those previously reported in the literature (although obtained with different methods), in particular regarding the role of the epithelium (for example https://doi.org/10.1186/s12989-020-00357-z)	We agree this is important. We've added a section to the discussion that contrasts our observations and methods with those of previous results showing direct villous uptake after bolus dosing by gavage. This section (Page 14 line 481) now reads: “ ... We show here that all features of this can be qualitatively and quantitatively recapitulated in a murine model, but achieving that requires exposure to human-relevant microbial 'danger signals'. Consistently, in this work, murine dosing with fgTiO₂ was undertaken in a manner that reflects real-life exposures (i.e., via the diet at relatively low levels for a long period of time). Interestingly, our findings contrast with a prior study, which used bolus dosing via oral gavage or surgical intervention in fasted animals, where direct fgTiO₂ uptake via the epithelium was reported. Future work may wish to consider the necessity of natural ingestion and concomitant normal microbial exposures, versus non-physiological dosing in carefully controlled environments, to recapitulate the human situation and address risk accurately for oral particle exposures. ” The field may need to consider the methods, observations and merits of both studies to

	successfully assess risk. With regard to particle delivery to the regular villous mucosa, rather than direct uptake, the revised work shows how a subset of the TiO₂-recipient cells of the Peyer's patch can migrate to the lamina propria after exposure to a microbial danger signal (new Figures 5-7). We also now show this exposure exists in human tissues (new Figure 1c/d). Please see our response to Comment 7 above for a short overview of these new findings. Many thanks.
Reviewer Three	
My background is purely computer science, image processing. I was asked by the editor to review the image processing of this manuscript. My understanding of the whole setting is the following. The goal is to detect and quantify the presence of fgTiO₂ additive in Peyer's patches (certain intestinal cells) in human tissue. The tissue was imaged using light and electron microscopies and X-ray spectroscopy to confirm that in the reflectance confocal microscopy the additive displays as bright foci, the brightness of which can be quantified subsequently. This confirmation has been reached qualitatively. Furthermore, two groups of mice, those fed also with fgTiO₂ and those that were not, were used to confirm this co-localization finding; and also to set up the image processing pipeline and, foremost, to confirm the scientific finding. The whole analysis is very much dependent on the outcomes of the image processing, making it a critical step in the whole study. Notice that tiny particles, which are hard to find for humans, are sought in the images and their area is computed. The overall area is rather small making every mis-detected pixel relatively influencing the overall computed outcome. I would, therefore, double-check every step of the pipeline, and would subject it to dedicated testing, and report this as well (to allow the readers to gain trust in the conducted measurements). This attitude is driven by my experience that image processing, including deep learning-based one, is in general very data sensitive, and requires fine-tuning to the particular data. Taking any pipeline, even when it has been published previously for the same or similar-looking data, does not necessarily guarantee any quality and accuracy of the calculations.	Thanks Vladimir for the careful review. It's extremely appreciated and has helped shape how we've carried out and reported the new biological experiments and image-analysis reporting when addressing the other reviewers' comments (described above). We entirely agree that it is important to subject the image processing to dedicated testing and to report this here as well. The new Supplementary Figure S2 now reports detailed accuracy testing of all cell segmentation pipelines using both mouse and human images by comparing the automated segmentation outputs to the results of manual annotations carried out by an experienced cell biologist using the intersection-over-union approach. The new Supplementary Figure 5 further provides examples of the raw reflectance signal, resulting circle-marker placements and then fgTiO₂-loaded vesicle segmentations achieved in both mouse and human images helping the reader assess and gain trust in the reliability of the process (please see our response to Comment 10 below). All three cell segmentation pipelines are displayed in the new Supplementary Notes 1-3 and are also available for download and testing from the (significantly revised) BioStudies submission, as is code and test-data demonstrating the thresholding and circle-marker placement.
(9) For example, Fig. 2b-d goes nicely in this direction by trying to question the sensitivity of the foci detection. Nevertheless, I haven't understood from the main text (L130-139), the figure, or its caption what is displayed in 2d, and why I see a bright spot in 2c (if it's supposed to be after the milling) in the white region, and why is there the yellow region.	The authors agree this wasn't sufficiently explained in the initial submission. Figure 2c is before milling and shows a scanning electron microscopy image overlaid onto the confocal-reflectance data with transparency so the underlying reflectant foci is still visible (the white spot). The white rectangle is caused by electron charging on the deposited platinum 'strap' that's laid down onto the tissue covering the region to be lifted out. The yellow box below this indicates the lower region to be milled away (the same process also happens above the

platinum strap, leaving behind the Pt-covered region to be lifted out (process shown, **Fig. B.**):

Fig. B. In situ lift-out process. (a) A platinum strap is laid down covering the region to be lifted out. (b/c) The lower then upper regions are milled out (d) the remaining platinum-covered lamella (indicated by red arrow) is attached to a manipulator and lifted out. *Images supplied by Andy Brown (co-author and electron microscopy lead).*

As such, **Figure 2D** shows a side-view of the lamella after lift-out. The platinum strap is now at the top of the image. This sample was then transferred into a scanning transmission electron microscope where (**Figure 2e-h**) high resolution analyses of the region in the red box reveal the single particle causing the reflectance signal. **We've re-written the Figure 2 legend to make this clear.** Thanks.

(10) Another step towards "gaining trust" in the detection would be to show a small number of insets with the original image without the translucent markers (before the markers were added) in the figures where the markers are used. Simply to see what the data looked like and to decide for myself if the marker was really where it should be.

This is a great suggestion! Please see the **new Supplementary Figure 5**. This shows the original reflectance data, subsequent circle-marker placements and the individual, fgTiO₂-loaded vesicle (TLV) segmentations for a range of examples taken from both the mouse and human images.

(11) In general, the Authors have done a great job of reporting how were images processed. I think I see the whole picture and understand (from the text, from the supplementary as well as from the EMBL-EBI-published code and data) what steps were taken and in which order. I have downloaded the EBI .zip file and familiarized myself with the available images. And I became a believer that the reported image processing could indeed work well on the data. However, the devil is in the details. And so I wrote down remarks that, if they would have been clarified in the manuscript right away, I would have been left without any doubts about the image processing used in this study. I think that the reported image processing is supporting the biological finding. That said, even if some details of the processing would have been carried out slightly differently, the overall outcome would continue supporting the biological claims. During the processing of the murine tissues, I understand and accept the plan that, given the fixed experiment and imaging conditions, the threshold level is determined on the non-fgTiO₂ images and then used in the fgTiO₂-containing images to detect it. However, when switching to human tissues, is it guaranteed that the fgTiO₂ signal appears in the same proportion compared to the rest of the tissue signal so that the same threshold level can reliably detect fgTiO₂. I am wondering if, for example, cell density is the same between the murine and human tissues (assuming the cell density may influence the propagation and thus strength of the wavelength).	The authors thank the reviewer for the positive comments about the reporting of the image processing. We agree that small changes to the processing would not affect the biological claims because this is really about relative differences: so long as all images contributing to an analysis are treated equally, the overall outcome should remain the same. The question regarding the validity of using the same threshold level in both mouse and human tissues is now addressed in the new Supplementary Figure 4. Plotting the reflectance intensity data for a series of 'background regions' that don't contain any foci (i.e., fgTiO₂ events) showed near-identical distributions in both mice and human tissues – supporting the use of single threshold that can be 'read across'. In part, this is likely aided by the optical sectioning imposed by the confocal microscopy technique because it means that the reflectance signal is not collected from the full depth of the tissue section but is instead precisely isolated to a thin plane (about 1 micron in z-depth) from the center of each section. Because of this, small differences (e.g., tissue section thicknesses or surface artifacts that are likely to occur between the mouse and human samples) won't contribute to the measured signal. Instead, the background signal in this volume will predominantly be caused by the actin cytoskeleton of the cells [4] – and thus driven by the cellular make-up of the tissue. This will be similar in the subepithelial dome tissue region in both mouse and human samples.
(12) Was there one (ideal) threshold that worked very well across all non-fgTiO₂ murine tissues? From the opposite side, I had a look at 'C4_section1.tif - C4_section1_3 - C=3.tif' file from the Figure 1 folder from the EMBL-EBI public data, and I see that the loci show themselves as clear (and high enough) intensity peaks rendering the (global) thresholding as indeed a viable technique. However, a range of threshold values was giving reasonable results, and with decreasing threshold was increasing the number of detected (and still rather reasonable looking) loci. The threshold interval is established from non-fgTiO₂ images. However, an interval of permissible thresholds (everything beyond a minimal threshold that "wipes out" everything) could be easily established for the non-fgTiO₂ images, opening a room to choose the final threshold, but this room could influence the number (and size) of the detected fgTiO₂ loci.	The reflectance signal in the non-fgTiO₂ animals was generally very consistent. This is because the optical sectioning imposed by the confocal microscope means that signal is only collected from a thin, ~ 1 micron section from the center of each tissue section thereby avoiding differences due to variation in section thickness or surface artifacts. A sense of this can be gained from looking at how similar the reflectance distributions are in background regions shown in the new Supplementary Figure 5. Whereas different thresholds could influence the number and size of the detected foci, what's critical is the relative difference - e.g., between the mouse and human images shown in Figure 3. For this reason, treating all images under comparison identically is more important than the specific threshold used. For this reason, image-sets for comparison were collected in a single imaging run using a microscope with diode-controlled laser excitation designed for stable, quantitative imaging. We've made sure this information/rationale is stated in the Methods (Page 20 Line 752 and Line 754).
(13) Considering Figs. 1 and 2, Supplementary Fig. 2, the lysomac cells were acquired as 3D stacks while the reflectance images seemed to be 2D, how was the matching slice from the 3D stack selected in	The 'stacks' of images shown in the original Supplementary Figures 1 and 2 where meant to indicate that multiple, 2-D images were analysed (not that 3-D stacks were used). We've now removed this as it is easy to mis-interpret! Thanks for highlighting this to us. Please

order to measure, e.g., the foci vs. full cell area (the cell dosimetry)?	see the revised Figures (now Supplementary Figures 1 and 3 also Figure 4 a/b/i/m). (They're also labelled '2-D input images' in the Figures to make sure this is clear).
(14) Since the confocal images were tile-scanned, with overlap, and then presumably registered and stitched, I understood that the lateral coverage of the reflectance images was obtained similarly, thus also by tile-scanning. How were the reflectance images registered before the stitching? Is it ruled out that the stitching process could introduce false positive foci?	Our apologies for the confusion, both the fluorescence information and the reflectance data were collected by the same microscope during each imaging run. The reflectance data is just another channel collected alongside the fluorescence information. We've made sure this information is now clearly stated at the top of the 'confocal microscopy' methods section (Page 20 Line 754). With respect to image registration, at each tile-position, images were collected with a (mechanically-achieved) 10% edge overlap then the manufacturer's software (Zen Black) was used to stitch the images together additionally guided by image similarity using the nuclei and actin channels. This important information was missing from the original manuscript. It's now been added to the methods section (Page 20 Line 756). Thank you for pointing this out. With respect to error and false-positive foci, because the tissue sections used were embedded in hard-set mountant (inhibiting movement of the specimen) and imaged in a temperature-controlled room (minimising thermal drift), it was extremely rare to encounter any errors with stitching process as only very minor shifts (i.e., stage movement errors) needed correcting. As such we are confident the stitching process did not introduce false-positive foci.
(15) How were the deep learning models trained? Especially, how were the training data prepared? If, for example, a network was classifying reflectance foci into three density classes Lo/Int/Hi, it must have been trained to do so. Was the classification done by one expert, or a consensus among experts (at least four to warrant any class is chosen at least two times)?	Our apologies for the confusion, the UNET deep learning model was only used to enable cell segmentation of the mouse lymphoid tissue images by providing a 'cell outline' probability image that more clearly delineated each cell's boundary. Please see the new Supplementary Figure 14 for a schematic explanation of this process. The training data for this model consisted of twelve lymphoid tissues images (each containing 2138 x 1900 annotated pixels covering ~16,000 cells prepared by an experienced cell biologist). This information is now included in the Methods on Page 24 Line 899. Moreover, all UNET code, training data, unseen test data and a pretrained model have now been added to the BioStudies project ('Mouse_lymphoid_tissue_UNET'). Regarding the low, int, hi mapping – this was not achieved by model classification. Instead, once each cell-object was segmented, the amount of fgTiO₂ delivered was measured by simply integrating the thresholded reflectance data cell-by-cell. This was achieved in the CellProfiler pipelines by a 'MeasureObjectIntensity' module. The integrated intensity information per cell was then binned into categories and cell-objects false-coloured accordingly to make the Figure visualisations using a simple MATLAB script. We've added a folder ('MATLAB_lo_int_hi_mapping') to the BioStudies deposition enabling download and running of this process. We've also significantly restructured / clarified the 'cell segmentation' section of the Methods to ensure this information is clearly relayed in the revised manuscript (Please see Page 23 Line 865 onwards.) Thanks.

(16) For the single-cell dosimetry measurements, the above set of questions applies also to the segmentation network that was used. Additionally, what level of values were there usually found in the segmentation probability maps? The Otsu thresholding cares about modes in the histogram, even if the modes are low (uncertain).

Please see our response to **Comment 15**. Deep learning models were only used for cell segmentation with dosimetry measures achieved simply by integrating fluorescence or reflectance information in these cell objects. We've thoroughly **revised the cell segmentation section of the Methods** to ensure this is clear (**Page 23 line 865 onwards**) and added new schematic diagrams to explain each cell segmentation process in **Supplementary Figures 13 – 16**.

A histogram of a typical probability image and its resultant segmentation are shown in **Fig. C** below. In the revised manuscript, we've included detailed cell segmentation accuracy assessment (new **Supplementary Figure 2**) for all cell segmentation pipelines ensuring they're performing in-line with our previously-published work [3].

Fig. C. Histogram of a typical probability map and typical probability map segmentation. (The black hole in the probability image represents a blood vessel).

(17) For the employed watershed, what seeds were used and how were they determined?

We've added a **new Supplementary Figure 13** that provides a clear schematic overview of the marker-controlled watershed process used for the mouse villous mucosa pipeline. In short, cell nuclei segmented from the images as 'PrimaryObjects' were used as the seeds to find the cell boundaries of each cell from the cytoskeletal (actin) staining. All of the cell segmentation pipelines are shown in **Supplementary Notes 1-3**. We've added four new schematic diagrams to the **Supplementary Figures 13-16** to ensure each is clearly explained. The running CellProfiler projects with data are also available for download from

	the BioStudies database (which has also been revised/improved, please see Fig. D in the next response, thanks).
(18) Note about code availability: The published code (from the EMBL-EBI site) is mostly a code to produce the figures themselves, not so much a code to conduct the analyses.	To address this, we've completely overhauled the EMBL-EBI BioStudies project submission in our revision. The new structure is shown below in Fig. D. The 'FIGURE_DATA' directory contains the raw data behind each of the Figures in the manuscript. The 'ANALYSIS_PIPELINES' directory provides running examples of the code used to conduct the analyses.  Fig. D. Files provided for download in the revised BioStudies submission accompanying the manuscript.
(19) PS: I think there is a systematic typo visulisation -> visualization.	Thanks for catching this. We've fixed throughout (and in the Supplementary Information file!).

References

- [1] European Medicines Agency. Final feedback from European Medicines Agency (EMA) to the EU Commission request to evaluate the impact of the removal of titanium dioxide from the list of authorised food additives on medicinal products. https://www.ema.europa.eu/en/documents/report/final-feedback-european-medicine-agency-ema-eu-commission-request-evaluate-impact-removal-titanium-dioxide-list-authorised-food-additives-medicinal-products_en.pdf - accessed March 26th, 2024.
- [2] Shreedhar VK, Kelsall BL, Neutra MR. Cholera toxin induces migration of dendritic cells from the subepithelial dome region to T- and B-cell areas of Peyer's patches. *Infect. Immun.* 2003, 71(1): 504-509.
- [3] *Supplementary Figure S3 in:* Wills JW, Robertson J, Summers HD, Minitier M, Barnes C, Hewitt RE, *et al.* Image-based cell profiling enables quantitative tissue microscopy in gastroenterology. *Cytometry A.* 2020, 97(12): 1222-1237.
- [4] Wills JW, Robertson J, Tourlomousis P, Gillis CMC, Barnes CM, Minitier M, Hewitt RE, Bryant CE, Summers HD, Powell JJ, Rees P. Label-free cell segmentation of diverse lymphoid tissues in 2D and 3D. *Cell Rep. Meth.* 2023.

Wills *et al.*, Immunocompetent Cell Targeting by Food-Additive Titanium Dioxide

Response to the reviewers' comments –

The authors thank the reviewers for the helpful feedback and thoughtful comments, which have helped us strengthen our manuscript. Utilising these, our revised submission **contains significant new data and we've restructured our work to better deliver the key take-home messages**. Importantly, dedicated Discussion sections now speak to the model-specificity of our findings and the implications for future work considering at-risk genotypes. This is unlocked through provision of a framework including an *in vivo* model with human-relevant dosimetry and identified target cells. We've also made all suggested edits to the Supplementary Figures supporting the image analysis work, ensuring the methodology is accessible to a broad audience.

Regarding new data, our revised submission now includes:

- Quantitative protein expression analyses (proximity ligation) for 92 cell regulatory / inflammatory endpoints in the wild-type (WT) and WT+TiO₂ mouse treatment groups at both +3 and +28-day timepoints (*i.e.*, in addition to the *Salmonella*- and TiO₂-*Salmonella* treatment groups included in our R1 submission).
- Single-cell immunofluorescence analyses establishing expression of key antigen presentation and co-stimulatory / inhibitory proteins (MHCII, CD80 and CD86) in the WT and WT+TiO₂ groups at both timepoints.
- Cell-type analyses using the markers MHCII, CD4 and cellular autofluorescence for the WT+TiO₂, +28-day timepoint pinning down the phenotype of the TiO₂-recipient cells from the sub-epithelial dome to the follicle base using high-resolution imaging of complete transverse sections of the Peyer's patch.
- The cellular autofluorescence data requested for the WT+TiO₂ group at both the +3 and +28-day time-points (with and without *Salmonella* exposure and concomitant fgTiO₂ migration).
- Pan-macrophage (IBA1) immunofluorescence labeling in the WT+TiO₂, *Salmonella*-exposed treatment group showing, as the reviewer anticipated, that some newly-recruited macrophages reside in the sub-epithelial dome of the Peyer's patch despite migration of the autofluorescence-positive cells.
- Spatial analyses of the number of TiO₂-loaded vesicles per cell and the amount of TiO₂ per vesicle broken down into the different functional zones of the Peyer's patch. This helpful suggestion enabled us to demonstrate that the vesicular inheritance process reported in our R1 submission specifically occurs below the sub-epithelial dome region in the mid- and basal- aspects of the patch.
- Single-cell image analytics more clearly demonstrating the tissue regions and ileal distribution of fgTiO₂ following *Salmonella* challenge.
- ICP-MS data establishing the levels of faecal titanium present at the different study timepoints / demonstrating that residual fgTiO₂ is still present at 16-week +3-day timepoint.

We believe the manuscript has been significantly improved through review and revision. It provides multiple, impactful contributions of global significance at an extremely timely point as regulatory bodies re-examine titanium dioxide and other oral exposures to persistent particles originating from pharmaceuticals, foods and consumer products.

A point-by-point response to all comments is provided below, with the positions of all changes corresponding to the marked-up manuscript **highlighted in yellow**.

Reviewer One	
Reviewer's comment	Response to the reviewer
Major comments	
I appreciate the authors' efforts to improve the quality and impact of their data. Throughout their revision, they carried out several experiments with specific objectives: 1/Confirming the identity of phagocytes responsible for capturing fgTiO₂ particles, which was crucial. Their effort revealed that fgTiO₂ were not only captured by macrophages (LysoMacs and pigment cells) but also by LysoDCs, thus addressing this point adequately. 2/Establishing the persistence of fgTiO₂ particles in Peyer's patches, especially in the subepithelial dome (SED) region, and their dispersion in the tissue post-infection. 3/Identifying any immunostimulatory activity of fgTiO₂ particles. However, despite these efforts, the newly acquired data are relatively minor, with the exception of identifying LysoDCs as an additional phagocyte subpopulation internalising TiO₂ particles and observing the persistence or dissemination of TiO₂ particles in the SED of PP at homeostasis or during infection, respectively. Without compelling functional assays, I maintain my belief that the authors should consider submitting their interesting yet descriptive work to a more specialized journal.	Thank you for your review and suggested experiments which enabled us to improve our submission. We politely disagree with your point about the 'fit' of our work with Nature Communications. Compelling findings of such widespread importance that transcend basic science to impact many disciplines (regulatory, toxicological, industrial, pharmaceutical, nutrition etc.) with great relevance to the broad field of persistent (nano)particle exposure should be reported in an equally compelling and important journal. Firstly, fgTiO₂ has become the exemplar for understanding intestinal exposures to (nano)particulates. Secondly, ongoing regulatory decisions are being pieced together from fragmented data and yet such decision-making impacts not just industry and the courts but the population as a whole – including future access to medicines as described in the Introduction of our manuscript. Our findings address critical, wide-ranging and time-bound issues: (i) what cells are actually targeted? (ii) how do these cells respond? (we now provide the new data requested for this) (iii) will fgTiO₂ augment an inflammatory response initiated by bacterial signalling in the target cells as is believed from in vitro studies? And (iv) here is a model of validated human relevance that can at long last be drawn upon to test key regulatory concerns (e.g., genotoxicity). Little of this would be addressed by 'functional' immunological assays and we have prioritised our work using state-of-the-art tools and carefully validated analyses to provide answers to the major questions in the field.
(1) Authors' first objective was addressed successfully, although for formal identification of cells (e.g., determining if all TiO₂+ cells with high autofluorescence express high CD4 but low MHCII levels?), it would have been preferable to display all markers and signal detections, i.e., TiO₂ particles, auto-fluorescence (AF), CD4 and MHCII, on the same cells instead of separately on different cells. Nevertheless, their complementary experiments seem to indicate that LysoDCs (AFint and CD4- and MHCIIhi) also capture TiO₂ particles.	The authors thank the reviewer for suggesting exploration of the potential for TiO₂ uptake by both LysoMacs and LysoDCs. We believe this finding has strengthened the revised work by demonstrating for the first time that immunocompetent cells with important roles in antimicrobial defence are targeted and become heavily loaded with fgTiO₂. This is particularly important to future utility of this work as it enables other questions regarding the safety of persistent oral particles to be precisely addressed using in vivo studies with human-relevant dosimetry and identified target cells (now discussed, Page 19 Line 602).

(2) Given the pivotal role that LysoDCs play in T cell priming, it is crucial to explore whether the uptake of TiO₂ particles influences the activation (e.g., surface expression of MHCII and co-stimulatory molecules after incubating sorted LysoDCs with TiO₂ particles, compared with known LysoDC stimuli) and priming capacity of LysoDCs (e.g., co-culture experiments with T cells). Despite the availability of well-established methods for such analysis, which have been previously published by different research groups and do not necessitate complex procedures, this aspect has not been investigated. Consequently, it remains challenging to conclusively state, as claimed in the abstract, that TiO₂ particles lack adjuvant effect. Incorporating these functional assessments would greatly enhance the study.	We greatly appreciate the rationale for approaching the functional effect of fgTiO₂ from a de novo immunological standpoint. However, beyond some early hypothesis around this question [1] there is not a body of experimental evidence suggesting that this is a priority question in the field. In contrast, the potential ability of fgTiO₂ to augment cellular inflammatory responses to bacterial components has been repeatedly claimed from in vitro studies [2-5] and whether this translates as a true in vivo finding is a priority question, in our opinion [6, 7]. Ahead of addressing this, two rather complex questions must be robustly answered. First, where do the particles accumulate at the per-cell level in vivo? Secondly, what is the methodology to co-expose those same cells to a primary bacterial signal of inflammatory effect that the fgTiO₂ could then augment (or not)? Our manuscript details these outcomes but we expand on the rationale for the model in our response to the reviewer's Comments 4 and 11, below. Notwithstanding, we have considered the reviewer's suggestion very carefully concerning a potential T cell adjuvant effect and taken advice from colleagues with expertise in the area. Collectively, it would not be trivial to extract and purify sufficient LysoDC cells from Peyer's patches, without activating them in the process, and to load them with particles / antigen in a way that mimics the in vivo setting and look at markers of adjuvanticity. 'If properly characterised and validated, it is probably the work of a PhD in an expert lab' was some feedback that we received. We did, however, address the reviewer's suggestion of looking at particle effects on MHCII and co-stimulatory molecules (CD80/86), specifically in the fgTiO₂⁺ cell population of the Peyer's patch SED (i.e., LysoDCs/LysoMacs). The data are shown below in Fig. A and reveal no measurable changes in response as a result of in vivo particle uptake. We have not, at this stage, included these results in the manuscript because, as noted above, our focus is very much on inflammatory augmentation. We are of course happy to do so if the reviewer / editor feels including these results would be helpful? We should finally note on this matter that we now recognise that our use of the word 'adjuvant' was misleading as this tacitly suggests amplification of an immune response (as the reviewer points out). We have corrected the text throughout to be clear that our experiment was about the ability (or not) of fgTiO₂ to initiate or augment inflammatory signalling at its site of accumulation and have removed the term 'adjuvant'.
---	--

Fig. A. *In situ*, single-cell expression analyses of MHCII and the co-stimulatory molecules CD80 and CD86 in the subepithelial dome of the Peyer's patch. (Top) Example images showing immunofluorescence labelling to quantify protein expression and reflectance microscopy to identify fgTiO₂ at the +3-day and +28-day timepoints. Translucent red circle-markers were placed on reflectant foci to aid visualization of fgTiO₂. **(Bottom)** Single-cell immunofluorescence distributions from wild-type (WT) mice fed with (WT + TiO₂) or without (WT) fgTiO₂ dietary supplementation (0.0625% w/w of diet). No significant differences in protein expression were observed across the two diet-groups.

(3) It is unfortunate that cellular markers were not used to assess the persistence of TiO₂ particles in the SED. Without such markers, it becomes challenging to definitely identify cells containing TiO₂+ vesicles at 28 days. Specifically, while the authors have noted a higher number of TiO₂+ vesicles per cell at 28 days, it remains unclear whether these cells are the same than those observed at 3 days. Given the different lifespans of LysoDCs and LysoMacs, the possibility of secondary capture by long-lived phagocytes should be envisaged. However, in the absence of markers it is impossible to draw any conclusion beyond a change in particle localisation post-infection.	In the revised work, we've considered the phenotype of these cells using high-resolution tilescreens of complete Peyer's patch transverse sections from the +28-day timepoint (presented, new Supplementary Figure 14). The work shows that the fgTiO₂ remains in autofluorescence-positive cells throughout their 'lifecycle' from the subepithelial dome (SED) to the base of the Peyer's patch. Along this journey, these cells exhibit a mixed CD4 and MHCII phenotype (as observed at the + 3 day timepoint) suggesting that a new population of long-lived phagocytes is not formed over time. With increasing distance from the SED, we did observe an increase in CD4 expression and a decrease in MHCII expression suggesting the particles ultimately become sequestered in cells with the longer-lived, LysoMac phenotype that exhibit increasingly reduced immunocompetence (e.g., in-line with the PD-L1 findings shown in Figure 3) (added to Results, Page 14 Line 429).
(4) The experimental approach used to assess the adjuvant activity of TiO₂ particles presents an issue as the influence of TiO₂ particles on the expression of inflammatory proteins is consistently examined alongside an infection using a non-virulent auxotrophic strain of Salmonella, known for its effectiveness as a vaccine. Considering the robust adjuvant activity demonstrated by this strain in numerous publications since the 1980s, it is unlikely that a specific effect of a potential adjuvant would be distinguishable in co-administration. Therefore, the relevance of such an experiment is limited, as multiple immune pathways are already activated by the attenuated bacteria via various PAMPs such as LPS, flagellin and bacterial DNA, among others. It would have been more pertinent to evaluate the effect of TiO₂ alone, in comparison to mice without TiO₂, within a model of tolerance versus classical immunisation.	The reviewer is correct that we should have first provided our data showing the effect of in vivo fgTiO₂ accumulation in the absence of Salmonella. This is now included and is referred to as 'initiating' an effect (in contrast to 'augmenting' with Salmonella – see below). No measurable difference compared to unexposed controls was identified (new Figure 5f and Results, Page 13 Line 391). Pietro Maestroni, with deep experience of Salmonella infection in the murine intestine, joined the team to help determine fgTiO₂-augmentation of an underlying inflammatory event in the intestine. The gut, lower gut especially, is hyporesponsive to bacterial fragments (i.e., does not elicit an inflammatory response). As such the attenuated ΔaroA-Salmonella strain was chosen as the 'initiating inflammatory factor' for several reasons. First, as the reviewer notes, it is a non-virulent auxotrophic strain so secondary inflammation, related to bacterial replication and clinical disease, should not compound the outcomes. Secondly, we reasoned that Salmonella, including this strain, would target the exact same cells as fgTiO₂ in the Peyer's patches. Thirdly, if co-localisation could be demonstrated then there was sufficient prior work to indicate that augmentation of immuno-inflammatory responses would be visible if they existed [8, 9] (Results, Page 12 Line 331). We therefore confirmed marked cellular co-accumulation (fgTiO₂ and Salmonella) and then used a highly sensitive and selective, non-biased, approach for multi-analyte assessment in the target tissue area at two time-points. We confirmed a modest (sic) but significant increase in Th1-associated inflammatory chemokines/cytokines for 4 out of 6 targets with ΔaroA-Salmonella alone versus bacterially-unexposed controls. But there was no suggestion that fgTiO₂ augmented this response or, indeed, any of the other 86 proteins (show, new Figure 5f/g and Supplementary Figures 12 and 13).
Specific and minor points	

(5) Figure 5f: this panel of the figure is very difficult to look at (multitude of very small panel with almost overlapping text). Perhaps the authors could consider selecting the most relevant molecules for display and relocating the others as supplementary data. Framed text is usually used to emphasized important elements of the figure; however, here, it is used to highlight analytes that are at least in part below the detection threshold. This is misleading. I would suggest to remove this frame.	The authors thank the reviewer for the helpful feedback. We now just show the key inflammation-associated protein targets in the revised Figure 5 with all other targets moved to the new Supplementary Figures 12 and 13. We've also removed the framed text.
(6) Figure 6: As mentioned above, incorporating cell markers such as CD11c, CD4, MHC-II and lysozyme would greatly improve the meaning of the data by providing an identity to TiO₂-loaded cells at day 28 compared to day 3.	We agree – and this is now presented in the new Supplementary Figure 14. Please see our response to Comment 3 above. Thanks.
(7) Figure 6a-b: I would suggest to add autofluorescence here to highlight the difference with panel L.	We agree this is an important addition. Please see the revised Figure 6 where this autofluorescence data is now included alongside the fgTiO₂ images right-most in panels a, b and k. The lack of fgTiO₂ and autofluorescence in the subepithelial dome after Salmonella challenge (Figure 6k) is now much clearer with this context. Many thanks.
(8) Figure 6c-d: It seems that there are already many TiO₂-loaded vesicles in the follicle at day 3, possibly even more than in the SED. Moreover, by day 28, these TLVs appear mostly located in the midpart of the follicles rather than at the basal part. To further elucidate these observations, quantifying the TiO₂-loaded vesicles according to specific regions, similar to the approach used in Figure 2n, would provide valuable quantitative data and enhance the analysis of the spatial distribution of these vesicles over time.	Thank you for this suggestion. We've broken this analysis up into three spatial zones ('SED-ROI', 'Mid-ROI', 'Base-ROI'; indicated in the new Figure 6 panels c and d). The analysis shows that whereas the dose of fgTiO₂ per vesicle remains remarkably consistent across all zones (new Figure 6h) the number of vesicles per cell increases in mid and base regions (new Figure 6e). This suggests that the vesicular inheritance process identified only comes into effect once cells leave the SED and ultimately leads to the formation of the heavily-loaded pigment cells at the follicle base (as is well-described in humans).
(9) Figure 7a: the quality of these images is rather poor making it challenging to visualize individual villi and even the lumen in the last picture. Improving the image quality would enhance clarity and facilitate a more precise interpretation of the data.	In the revised Figure 7A we've improved the resolution and added labels to better indicate the subepithelial dome (SED), lumen and villous mucous (VM) regions in each tile scan. We've also included enlarged versions of these three analyses alongside the fluorescently-counterstained images to provide improved histological context in the new Supplementary Figure 15.
(10) Result section: I. 264: ref 27, Monack's lab work is anterior to the characterization of LysoDC's in the early 2010s.	Reference removed (Page 12, Line 333). Thanks for pointing this out.
(11) I. 265-266: Several studies spanning several decades have demonstrated not "modest" but rather robust adjuvant activity of the attenuated Salmonella strain. Although this strain does not induce disease as it is rapidly eliminated, it is known to elicit potent humoral and cellular immune responses.	Please see our response to Comment 4 above, thanks.

(12) l. 303: this sentence may be misleading as it could imply the absence of macrophages in the SED at day 28, which is unlikely. It would be beneficial to stain macrophages (e.g., MerTK and Lysozyme) in this context to provide clarity. The most plausible explanation is that the renewal of macrophages following infection likely decreased their autofluorescence, as newly recruited macrophages may have a reduced phagocytic history compared to those they replaced.

We understand the reviewer's point. In the text, we've made sure to specifically describe the absence of autofluorescence-positive cells (and not the absence of macrophages) **Page 14, Line 436**. This section now reads:

SEDs were almost entirely devoid of fgTiO₂ and autofluorescence-positive cells at +28 days (Figure 6k), while some cells of the regular villous lamina propria, as well as of the basal aspect of the Peyer's patch, were now positive for fgTiO₂ (Figure 7a-c; enlarged display with counterstaining context, Supplementary Figure 15).

During the revisions, we also labelled a +28-day section from the WT+TiO₂+*Salmonella* treatment group with the pan-macrophage marker [10, 11] IBA1 (shown, **Fig. B, below**). This indicates the presence of newly-recruited macrophages in the SED as the reviewer anticipated. **We can add this data to the Supplementary Information if the reviewer thinks it helpful?** (but it was just our intention to point out the lack of autofluorescence-positive cells and not macrophages in general).

Fig. B. IBA1 immunofluorescence labelling indicates the presence of newly-recruited macrophages in the mouse SED after *Salmonella* challenge. Confocal microscopy data collected from a WT+TiO₂+*Salmonella* treated mouse at the +28-day timepoint.

(13) l. 310-l. 315: without labeling with LysoDC and LysoMac specific markers and excluding villus lamina propria macrophages, there is insufficient evidence to support this scenario. Therefore, it would be

In the revised manuscript we've reworked and moved this section to the Discussion **Page 16, Line 513**. Many thanks.

more appropriate to move these sentences to the discussion section. Additionally, recent studies have demonstrated that indeed stimulated LysoDCs migrate, albeit to the IFRs rather than the villus lamina propria.	
Reviewer 2	
The submitted paper analyzes the fate of food-grade titanium dioxide in the intestine of mice identifying, in the subepithelial dome region of Peyer's patches, LysoMAC and LysoDC cells as the final destination of this additive. This observation obtained in mice was confirmed by the analysis of human surgical samples, where titanium dioxide was localized in the same cell types. Due to the active role of DC cells in the immune response, the authors evaluated if the presence of titanium dioxide was able to alter the response to attenuated ΔaroA-Salmonella, which colocalized in the same cells. The assessment of the level of several proteins involved in immunity and inflammation did not show any difference in the animal previously treated with titanium dioxide compared to those on normal chow. The data presented in the paper are providing convincing evidence regarding the localization of the titanium dioxide particles in the Peyer's patches.	Thank you for reviewing the work and for the positive feedback regarding the provision of convincing evidence regarding the localisation of the fgTiO₂ in intestinal tissues. Your Comments below were also extremely helpful. We have added new experimental data and carefully restructured our Discussion to address your points (please see our responses below / a summary of the new data at the top of the document).
(14) However in the discussion the authors should underline that the lack of effect of titanium dioxide on the immune response is strictly limited al model employed in this study.	We agree this is important. In our revised Discussion, we've now included a paragraph that describes the potential model-specificity of our findings relative to previous work. We've also added a dedicated section discussing the need for future studies considering at-risk genotypes. Revised Discussion, Page 15, Line 480 discusses the important differences between this work using a murine oral-exposure model relative to previous studies using oral gavage: Consistently, in this work, murine dosing with fgTiO₂ was undertaken in a manner that reflects real-life exposures (i.e., via the diet at relatively low levels for a long period of time). Our findings contrast with a prior study, which used bolus dosing via oral gavage or surgical intervention in fasted animals, where direct fgTiO₂ uptake via the epithelium was reported³⁹. Future work may wish to consider the necessity of natural ingestion and concomitant normal microbial exposures, versus non-physiological dosing in specific pathogen free environments, to recapitulate the human situation and address risk accurately for oral particle exposures. Revised Discussion, Page 17, Line 558 then speaks to the potential role of fgTiO₂ in intestinal pathology and the need for future studies considering at-risk genotypes

	(including the potential implications for Crohn's disease): Role in intestinal pathology. [...] Notwithstanding, the potential for fgTiO₂- induction or augmentation of intestinal pathology still deserves scrutiny. First, in subjects with Crohn's disease, fine-particle-targeted macrophages of the Peyer's patch do not express PD-L1²⁵. Interestingly, Crohn's disease is a relatively modern inflammatory disorder and there is good evidence that it may initiate in the Peyer's patches⁴². The most commonly-associated single gene mutation with the disease is NOD2 and there is in vitro evidence that when NOD2 is perturbed fgTiO₂ augments enhanced inflammatory activity with microbial fragments¹. So it should now be carefully considered what impact the ingestion of fgTiO₂ has on at-risk genotypes and, especially, where NOD2 functionality is diminished or removed. Secondly, it is important to note that conclusions from this work refer to the distal small intestine where there is uptake of fgTiO₂. As noted previously¹², however, it does not shed light on potential effects of fgTiO₂ where it is not absorbed such as in the large intestine. Direct luminal effects (on microbiome or apical enterocytes for example) cannot be precluded, especially given that this is where exposure will be by far the greatest. Indeed, both Urrutia-Ortega et al.,⁴³ and Bettini et al.,⁴⁴ have provided evidence for direct, pro-tumorigenic effects of fgTiO₂ on the colon which deserves further attention¹².
(15) In addition, in their experimental design the administration of titanium was suspended before that of the ΔaroA-Salmonella. Due to the fact that titanium dioxide is a food additive, it would be quite important to assess if its interaction with food components can alter their presentation to the intestinal immune system.	This is a great question and one that we considered at length during the experimental design stage. On the one hand, we did not wish to adsorb the administered bacteria to the fgTiO₂ particles and simply wash the bacteria out of the system and into the faeces. This is well known in similar systems [12] and could have potentially indicated an artefactual advantage of fgTiO₂ simply due to co-administration of the two (particles + bacteria). In 'real life' it is likely that oral exposure to pathogenic bacteria is an occasional event and, when it does occur, is not often in the immediate presence of food-grade TiO₂ which is only found in distinct processed foods, pharmaceuticals and nutraceuticals. In contrast, prior studies suggest that the large majority of the (Western) population is exposed to oral fgTiO₂ at some stage during a 24-48 hour window [13]. In our revised manuscript, we include ICP-MS studies showing the levels of faecal titanium at the different mouse-study timepoints (new Supplementary Figure 17). The data show that it takes >3 days to flush fgTiO₂ through the intestine in entirety so luminal interactions with bacteria should be allowed for in any experimental design (if indeed it will occur). We therefore marginally separated the administrations of bacteria and fgTiO₂ to (i) prevent artefactual adsorption due to co-administration (ii) enable luminal interactions between gavaged bacteria and residual luminal particles and (iii) ensure that food also was in the lumen under these circumstances (as the reviewer notes, this is important for a fully physiological assessment). The rationale for these choices is now included in the revised Method (Page 21, Line 467).

(16) Moreover, although the authors did not show any accumulation of titanium dioxide in the epithelium they cannot exclude any functional damage to enterocytes, and this should also be taken into account. Last but not least, the lack of additive effect cannot be excluded in condition affecting the intestine, such as IBD or celiac disease.	We entirely agree; this Comment exemplifies the importance of the interplay between reviewers and authors. The Discussion deserves better balance which we've tried to address in this revision. Importantly, we've included a section that covers (i) the need for future work in at-risk genotypes and (ii) the potential for direct luminal effects: Revised Discussion, Page 17, Line 558: Role in intestinal pathology. [...] Notwithstanding, the potential for fgTiO₂- induction or augmentation of intestinal pathology still deserves scrutiny. First, in subjects with Crohn's disease, fine-particle-targeted macrophages of the Peyer's patch do not express PD-L1²⁵. Interestingly, Crohn's disease is a relatively modern inflammatory disorder and there is good evidence that it may initiate in the Peyer's patches⁴². The most commonly-associated single gene mutation with the disease is NOD2 and there is in vitro evidence that when NOD2 is perturbed fgTiO₂ augments enhanced inflammatory activity with microbial fragments¹. So it should now be carefully considered what impact the ingestion of fgTiO₂ has on at-risk genotypes and, especially, where NOD2 functionality is diminished or removed. Secondly, it is important to note that conclusions from this work refer to the distal small intestine where there is uptake of fgTiO₂. As noted previously¹², however, it does not shed light on potential effects of fgTiO₂ where it is not absorbed such as in the large intestine. Direct luminal effects (on microbiome or apical enterocytes for example) cannot be precluded, especially given that this is where exposure will be by far the greatest. Indeed, both Urrutia-Ortega et al.,⁴³ and Bettini et al.,⁴⁴ have provided evidence for direct, pro-tumorigenic effects of fgTiO₂ on the colon which deserves further attention¹².
Reviewer 3	
I would like to thank the Authors for carefully and patiently answering all my questions, and I very much appreciate that they conducted additional experiments and analyses also towards strengthening the description of the image processing and analysis parts, and that they cared to describe everything in the manuscript. Also thanks for the information and all three figures in the rebuttal letter. The new insights into the pipelines and pokes into the intermediate results make the whole study a lot stronger for me. I have to say that the newly reported values are very good, and convincing. For example, the new Supplementary Figures 2 and 5 are both nicely illustrating the data, the image processing problem, and the typical outcomes. Besides, Supp. Fig. 2 shows additionally very good segmentation accuracy!	The authors are grateful for the thorough review, which enabled us to markedly improve the dissemination / accessibility of the image analysis method. It's great to hear that the approach is convincing and that the methodological concerns have been resolved. Thank you for your help, which has significantly improved our work.

Regarding the settings of the threshold: It is true that previously I haven't seen the connection that the given biological-physical-optical setup could warrant that the distributions of image intensities across both human and mouse images can be very similar, and thus the threshold value established in the first group of images can be carried over to the second group of images. I thank the Authors for reminding me, and especially thank them for providing the new Supplementary Figure 4 that demonstrates the situation perfectly. (And there's no need to quantify the observation any further.) These two topics above were previously my two greatest concerns about the methodology, which are now cleared away. The updated descriptions of the image processing, in the Methods section, are now clearer to me too because they contain now more of a technical context and reasoning to justify the steps taken. I find this new part of the manuscript to be far more easily reproducible than what it was before. And I again thank the Authors for their effort on this. The publicly available data on the BioImage Archive are findable, accessible, interoperable, and seem reusable (upon random digs into the source codes and images), very nicely organized now (thank you again).	
Minor comments	
(17) A very minor remark would still be left, though. In the new Supp. Fig. 13, the white rectangle in panel c) is not echoed elsewhere in that figure (and is also not mentioned in the caption); could it be because of the white frame on a white background? And there's also maybe(?) a mismatch in the descriptions as the b) segmentation of nuclei looks more like a segmentation of a,top) the actin channel rather than a,bottom) the nuclei channel; the descriptions in panel a) are probably swapped. And could the segments boundaries be drawn in e.g. green (not black) in panel d) to facilitate qualitative demonstration of the watershed performance? (It's okay and normal if the performance is not superb everywhere.) Similarly, the segments' contours in [Supp. Fig. 14 c), two right most] and [Supp. Fig. 15 f,g] are suggested black when the actual contours are white; could they be e.g. green like they already are around?	Thanks for these excellent corrections / improvements, we've made them all: In the revised Supplementary Figure 18 the rectangle in panel (c) is now shown in grey and echoed in the outline of panel (d). We've corrected the mismatched descriptions in panel (a) and changed the segmentation outlines to green in panel (d). In the revised Supplementary Figures 19 and 20 the segmentation boundaries in panels (c), (f) and (g) are now shown in green with matching green text-labels.

Remarks on code availability	
(18) I haven't checked all the code that's now available, I checked only randomly some files. But I can conclude that the code+data is well organized, and the source code is understandable. It is a mixture of Matlab, Python, and CellProfiler scripts, but that's okay. I haven't tried to execute anything.	Thank you for the random spot-checks on the files – it's appreciated – and great to hear that the source code is understandable.

REFERENCES

1. Powell, J.J., V. Thoree, and L.C. Pele, *Dietary microparticles and their impact on tolerance and immune responsiveness of the gastrointestinal tract*. Br J Nutr, 2007. **98 Suppl 1**(Suppl 1): p. S59-63.
2. Evans, S.M., et al., *The role of dietary microparticles and calcium in apoptosis and interleukin-1beta release of intestinal macrophages*. Gastroenterology, 2002. **123**(5): p. 1543-53.
3. Butler, M., et al., *Dietary microparticles implicated in Crohn's disease can impair macrophage phagocytic activity and act as adjuvants in the presence of bacterial stimuli*. Inflamm Res, 2007. **56**(9): p. 353-61.
4. Ashwood, P., R.P. Thompson, and J.J. Powell, *Fine particles that adsorb lipopolysaccharide via bridging calcium cations may mimic bacterial pathogenicity towards cells*. Exp Biol Med (Maywood), 2007. **232**(1): p. 107-17.
5. Riedle, S., et al., *Pro-inflammatory adjuvant properties of pigment-grade titanium dioxide particles are augmented by a genotype that potentiates interleukin 1 β processing*. Part Fibre Toxicol, 2017. **14**(1): p. 51.
6. Hewitt, R.E., H.F. Chappell, and J.J. Powell, *Small and dangerous? Potential toxicity mechanisms of common exposure particles and nanoparticles*. Curr Opin Toxicol, 2020. **19**: p. 93-98.
7. da Silva, A.B., et al., *Gastrointestinal Absorption and Toxicity of Nanoparticles and Microparticles: Myth, Reality and Pitfalls explored through Titanium Dioxide*. Curr Opin Toxicol, 2020. **19**: p. 112-120.
8. Agorio, C., et al., *Live attenuated Salmonella as a vector for oral cytokine gene therapy in melanoma*. J Gene Med, 2007. **9**(5): p. 416-23.
9. Eom, J.S., et al., *Enhancement of host immune responses by oral vaccination to Salmonella enterica serovar Typhimurium harboring both FliC and FljB flagella*. PLoS One, 2013. **8**(9): p. e74850.
10. Yip, J.L.K., et al., *Macrophage regulation of the "second brain": CD163 intestinal macrophages interact with inhibitory interneurons to regulate colonic motility - evidence from the Cx3cr1-Dtr rat model*. Front Immunol, 2023. **14**: p. 1269890.
11. Chiaranunt, P., et al., *Beyond Immunity: Underappreciated Functions of Intestinal Macrophages*. Front Immunol, 2021. **12**: p. 749708.
12. Siemer, S., et al., *Nanosized food additives impact beneficial and pathogenic bacteria in the human gut: a simulated gastrointestinal study*. NPJ Sci Food, 2018. **2**(1): p. 22.
13. Lomer, M.C., et al., *Dietary sources of inorganic microparticles and their intake in healthy subjects and patients with Crohn's disease*. Br J Nutr, 2004. **92**(6): p. 947-55.

Wills *et al.*, Immunocompetent Cell Targeting by Food-Additive Titanium Dioxide

Response to the reviewer's comments –

Reviewer 1	
Reviewer's comment	Response to the reviewer
As this is the second round of review, I will only respond to the points I previously raised that have not been addressed and the reasons provided by the authors for this, or for which I still have concerns. Other points have been adequately addressed by the authors and I thank them for that.	Thank you for the review and for the positive comments regarding the other work completed.
Authors' response: Notwithstanding, we have considered the reviewer's suggestion very carefully concerning a potential T cell adjuvant effect and taken advice from colleagues with expertise in the area. Collectively, it would not be trivial to extract and purify sufficient LysoDC cells from Peyer's patches, without activating them in the process, and to load them with particles / antigen in a way that mimics the in vivo setting and look at markers of adjuvanticity. 'If properly characterised and validated, it is probably the work of a PhD in an expert lab' was some feedback that we received. (1) My comments on this response: I understand that if none of the authors have expertise in in vitro culture of phagocytes, such experiments may seem challenging. However, other straightforward functional tests could have been performed instead, such as DTH (delayed-type hypersensitivity) response tests to determine at least whether TiO₂ can induce a loss of oral tolerance. This is a critical point to evaluate its innocuity. These types of tests do not require cell extraction but just involve measuring footpad or ear swelling after mice have been fed several times with an antigen (e.g., OVA), immunized with the same antigen and then challenged subcutaneously in the footpad or ear (e.g., PMID: 1653388).	The reviewer continues to make a cogent case for investigations around fgTiO₂ as a potential T cell adjuvant. We appreciate this advice and have used their suggestions to build this into our revised Discussion as a third priority area that can now be addressed using the human-relevant murine model provided. Specifically (Page 18 Line 447), this reads: Secondly, persistent particulates in the size range of fgTiO₂ are recognised as potential immuno-adjuvants, notably in skewing the magnitude and direction of T cell proliferative responses [1]. Studies reported herein, demonstrating no measurable augmentation of intestinal inflammatory responses by fgTiO₂, do not preclude a separate immuno-adjuvant role for fgTiO₂. Ex-vivo studies, using fgTiO₂- and/or antigen- loaded LysoMacs and LysoDCs to stimulate T cell proliferation and activation should be considered. In vivo, oral adjuvant effects of fgTiO₂, with a co-ingested neo-antigen, could similarly be investigated as, for example, was previously reported for lactoferrin with delayed type hypersensitivity as a read out [2].
Authors' response: We did, however, address the reviewer's suggestion of looking at particle effects on MHCII and co-stimulatory molecules (CD80/86), specifically in the fgTiO₂ + cell population of the Peyer's patch SED (i.e., LysoDCs/LysoMacs). The data are shown below in Fig. A and reveal no measurable changes in response as a result of in vivo particle uptake. We have not, at this stage, included these results in the manuscript because, as noted above, our focus is very much on inflammatory augmentation. We are of course happy to do so if the reviewer / editor feels including these results would be	We appreciate the reviewer's advice and have not included this Figure. The BioStudies submission and antibody information Table (Supplementary Table 2) have been updated to reflect this. Thanks.

helpful? (2) My comments on this response: Unfortunately, changes in MHCII, CD40, CD80 and CD86 surface expression need to be measured by flow cytometry to achieve sufficient sensitivity. Therefore, I suggest excluding this figure, as it is misleading and does not effectively convey the intended information.	
Authors' response: In the revised Figure 7A we've improved the resolution and added labels to better indicate the subepithelial dome (SED), lumen and villous mucous (VM) regions in each tilescan. We've also included enlarged versions of these three analyses alongside the fluorescently-counterstained images to provide improved histological context in the new Supplementary Figure 15. (3) My comments on this response: It is necessary to include the actin staining shown in Supplementary Figure 15 in the main figure, as it provides valuable insight into the tissue morphology. The authors should label all SEDs, not just one (image 2 and 3 contain two SEDs each), and use actin staining (brush border) to accurately delineate the follicle-associated epithelium (FAE). Notably, the FAE border in image 2 is incorrectly marked. Additionally, the term "basal aspect" in the main text is misleading. The authors should adhere to Peyer's patch (PP) nomenclature. Specifically, the PP regions located in between follicles (and SEDs) are termed interfollicular regions (IFRs). These are the regions where part of TiO2-loaded cells accumulates upon Salmonella infection, distinct from pigment cells observed at the base of the IFRs and follicles. IFRs are where naïve T cells are located and primed, and where LysoDCs, but not LysoMacs, migrate upon stimulation (PMID: 37580335 and PMID: 32268097). This should be clearly mentioned in the manuscript.	We thank the reviewer for the helpful input. The revised Figure 7 now incorporates the actin staining previously shown in Supplementary Figure 15. All SEDs are labelled and we've revised the Figure legend, Results (Page 15 Line 366) and Discussion (Page 16 Line 406) to remove the term basal aspect, point out migration to the IFRs and incorporate the suggested references.
Minor points:	
(4) L97-99: Given the topic of the study, the authors should clearly mention here that LysoDCs and LysoMacs are specialized for the uptake of dead cells, particulate antigens and pathogenic bacteria, in addition to their role in anti-microbial defense. This is particularly relevant since TiO2 uptake is the main focus of the manuscript.	We've added the suggested text to the Introduction (Page 5 Line 105). Many thanks.
(5) L269-272: The phrase "resulting in a modest immuno-inflammatory response (unlike bacterial fragments, to which the intestine is uniquely tolerant)" should be removed, as the immune response to this strain is strong enough to induce long-term protection against the virulent strain, and thus cannot be considered "modest".	We've removed this phrase from the Results (Page 12 Line 278). Thanks.

(6) L329-330: Rather than depleting the SED region of previously absorbed plastic microparticles, oral danger signals, such as Salmonella, induce the migration of antigen-loaded LysoDCs from the SED to the IFRs (PMID: 12496201, PMID: 37580335 and PMID: 32268097). This needs to be specified.	We've specified this in the Results (Page 15 Line 364) and incorporated the helpful references. Thanks.
(7) L333-334 and 371-372: There are several hypotheses that could explain the presence of TiO₂ in the villi during Salmonella infection. Regardless of the type of stimulation, including Salmonella infection, LysoDCs have never been observed in the villus lamina propria. Therefore, if the authors wish to support the hypothesis of LysoDC migration into villi, they must at least demonstrate that these villus TiO₂-loaded cells exhibit the phenotype of LysoDCs. This includes showing lysozyme expression and the absence of CD4, as well as the lack of the villus macrophages markers CD64 and F4/80. Autofluorescence alone is characteristic of all macrophages and cannot be used to define LysoDCs, which are actually less autofluorescent than LysoMacs. If such evidence is not provided, these lines should be removed, as it is not supported by the data.	Thank you for the input. We've removed the lines suggesting LysoDC migration to the villous mucosa (Page 15 Line 368).
Reviewer 2	
No further comments	
Reviewer 3	
No further comments	

REFERENCES:

1. Pele, L.C., et al., *Peripheral blood mononuclear cell proliferative responses to soluble and particulate heat shock protein 65 in health and inflammatory bowel disease*. Inflamm Res, 2007. **56**(4): p. 143-8.
2. Zimecki, M., et al., *Lactoferrin as an Adjuvant for the Generation of Delayed Type Hypersensitivity to Orally Administered Antigen*. Ann Clin Lab Sci, 2021. **51**(3): p. 359-367.